# Improving Denoising Diffusion with Efficient Conditional Entropy Reduction

## Abstract

Diffusion models (DMs) have achieved significant success in generative modeling, but their iterative denoising process is computationally expensive. Training-free samplers, such as DPM-Solver, accelerate this process through gradient estimation-based numerical iterations. However, the mechanisms behind this acceleration remain insufficiently understood. In this paper, we demonstrate gradient estimation-based iterations enhance the denoising process by effectively *reducing the conditional entropy* of reverse transition distribution. Building on this analysis, we introduce streamlined denoising iterations for DMs that optimize conditional entropy in score-integral estimation to improve the denoising iterations. Experiments on benchmark pre-trained models validate our theoretical insights, demonstrating that numerical iterations based on conditional entropy reduction improve the reverse denoising diffusion process of DMs. The code will be available.

## 1 Introduction

It is well established that diffusion models (DMs) (Sohl-Dickstein et al., 2015; Ho et al., 2020; Song et al., 2021b) have achieved significant success across various generative tasks, including image synthesis and editing (Dhariwal & Nichol, 2021; Meng et al., 2022), text-to-image synthesis (Ramesh et al., 2022), voice synthesis (Chen et al., 2021), and video generation (Ho et al., 2022). DMs consist of a forward diffusion process and a reverse denoising diffusion process. In the forward process, Gaussian noise is progressively injected into the data, perturbing the data distribution to collapse towards a standard Gaussian distribution by increasing conditional entropy. During training, the neural network is tasked with learning to reverse this process by minimizing the loss between the predicted and injected noise. Once the model is well-trained, high-quality samples can be synthesized by simulating the reverse-time denoising process associated with the forward noise-adding process.

However, a key limitation of DMs is the slow sequential nature of their iterative denoising process (Song et al., 2021a). To overcome this, training-free methods aim to accelerate denoising process by efficient numerical iterative algorithms without requiring additional training or costly optimization. Many of these methods focus on reformulating the denoising process as the solution of an ODE, allowing for accelerated sampling through numerical techniques. Such examples include PNDM (Liu et al., 2022), EDM Karras et al. (2022), DPM-Solver (Lu et al., 2022a), DEIS (Zhang & Chen, 2023), UniPC (Zhao et al., 2024), and DPM-Solver-v3 (Zheng et al., 2023a).

Despite the success of these numerical discretization techniques, the underlying mechanisms driving their acceleration remain inadequately understood. In particular, the reasons why iterations with similar orders of convergence result in varying levels of acceleration are not well explored. To address this gap, we reexamine the principles driving the accelerated denoising process. Our conditional entropy-based analysis reveals that effective iterations systematically reduce the conditional entropy of denoising transition distributions at each step, thereby directly contributing to a faster denoising process. This insight clarifies the mechanisms of gradient-based acceleration and provides a foundation for designing efficient denoising algorithms. Our main contributions are as follows:

- We introduce a novel perspective on *entropy reduction* in denoising diffusion of DMs, demonstrating that *gradient estimation-based iterations significantly accelerate the denoising process by effectively **reducing conditional entropy***. Our theoretical analysis further reveals that denoising iterations *using data-prediction parameterization are more effective* than those using noise-prediction parameterization in minimizing conditional entropy.

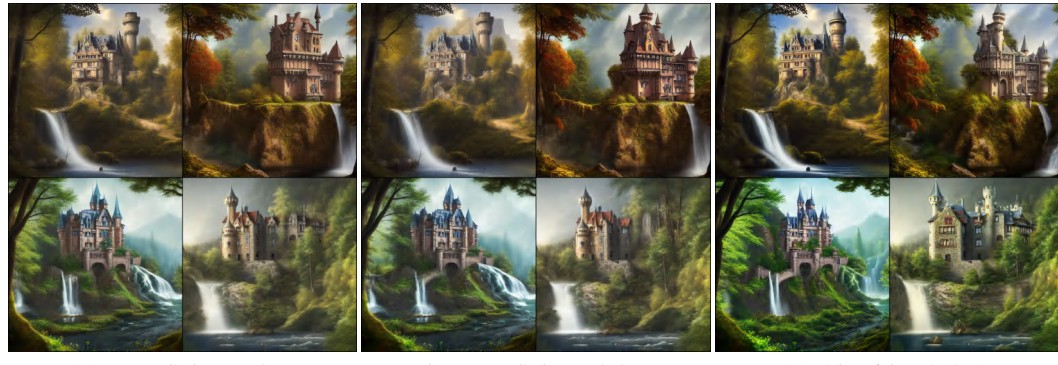

| (a) DPM-Solver++2m. | (b) DPM-Solver-v3-2m. | (c) Algorithm 1 (2m). |

Figure 1: Random samples from Stable-Diffusion Rombach et al. (2022) with a classifier-free guidance scale 7.5, using 10 number of function evaluations (NFE) and text prompt "*A beautiful castle beside a waterfall in the woods, by Josef Thoma, matte painting, trending on artstation HQ*".

- Building on our theoretical insights, we propose a denoising iteration method focused on efficient reducing conditional entropy in DMs. Unlike existing training-free methods, our approach improves the denoising process by lowering variance-driven conditional entropy during gradient-based iterations, which provides a simple yet effective improvement.
- Experiments on benchmark pre-trained models in both pixel and latent spaces validate our theoretical insights and demonstrate that our proposed method not only matches but often improves the reverse denoising diffusion process in DMs.

## 2 BACKGROUND

Diffusion models (DMs) define a Markov sequence $\{\boldsymbol{x}_t\}_{t\in[0,T]}$ in the forward process, starting with $\boldsymbol{x}_0$, where $\boldsymbol{x}_0 \in \mathbb{R}^d$ is drawn from the clean data distribution $q_0(\boldsymbol{x}_0)$. This sequence is pushed forward with increasing entropy until it approaches a standard Gaussian distribution via the transition kernel: $q_t(\boldsymbol{x}_t \mid \boldsymbol{x}_0) = \mathcal{N}\left(\boldsymbol{x}_t; \alpha_t \boldsymbol{x}_0, \sigma_t^2 \boldsymbol{I}\right)$, where $\sigma_t$ are smooth monotonic scalar functions w.r.t $t$. In DMs, $\alpha_t$ and $\sigma_t$ are called as the noise schedules, $\alpha_t^2/\sigma_t^2$ is called the signal-to-noise ratio (SNR) function. This transition kernel can be reformulated as the equivalent stochastic differential equation (SDE):

$$\mathrm{d}\boldsymbol{x}_t = f(t)\boldsymbol{x}_t \, \mathrm{d}t + g(t)\mathrm{d}\boldsymbol{\omega}_t, \ \boldsymbol{x}_0 \sim q_0(\boldsymbol{x}_0), \tag{2.1}$$

where $\boldsymbol{\omega}_t$ denotes a standard Wiener process, $f(t) := \frac{\mathrm{d}\log\alpha_t}{\mathrm{d}t}$, $g^2(t) := \frac{\mathrm{d}\sigma_t^2}{\mathrm{d}t} - 2\frac{\mathrm{d}\log\alpha_t}{\mathrm{d}t}\sigma_t^2$ (Kingma et al., 2021). The reverse-time SDE of above forward diffusion process can be written as:

$$\mathrm{d}\boldsymbol{x}_t = \left[f(t)\boldsymbol{x}_t - g^2(t)\nabla_{\boldsymbol{x}}\log q_t(\boldsymbol{x}_t)\right]\mathrm{d}t + g(t)\mathrm{d}\overline{\boldsymbol{\omega}}_t, \quad \boldsymbol{x}_T \sim q_T(\boldsymbol{x}_T), \tag{2.2}$$

where $\overline{\boldsymbol{\omega}}_t$ represents another standard Wiener process. In score-based models (Song et al., 2021b), the diffusion (or probability flow) ordinary differential equation (ODE) used for efficient sampling is derived from the Fokker-Planck evolution equation of the probability density function as follows:

$$\frac{\mathrm{d}\boldsymbol{x}_t}{\mathrm{d}t} = f(t)\boldsymbol{x}_t - \frac{1}{2}g^2(t)\nabla_{\boldsymbol{x}}\log q_t(\boldsymbol{x}_t), \tag{2.3}$$

where the marginal distribution $q_t(\boldsymbol{x}_t)$ of $\boldsymbol{x}_t$ is equivalent to that of $\boldsymbol{x}_t$ in the SDE presented by Eq. (2.2). To train DMs, following the practiced in DDPM Ho et al. (2020), a neural network $\epsilon_\theta(\boldsymbol{x}_t, t)$ is parameterized to predict the noise $\epsilon$ by minimizing the expectation of mean squared error as follows:

$$\mathbb{E}_{\boldsymbol{x}_0\sim q_0(\boldsymbol{x}_0), \ \epsilon\sim\mathcal{N}(\boldsymbol{0},\boldsymbol{I}), \ t\sim\mathcal{U}(0,T)}\left[w(t)\|\epsilon_\theta(\alpha_t\boldsymbol{x}_0 + \sigma_t\epsilon, t) - \epsilon\|_2^2\right], \tag{2.4}$$

where $\alpha_t^2 + \sigma_t^2 = 1$, $w(t)$ is a weighting function that depneds on the evolution time $t$. By substituting the trained noise prediction model $\epsilon_\theta(\boldsymbol{x}_t, t)$ with the scaled score function: $-\sigma_t\nabla_{\boldsymbol{x}}\log q_t(\boldsymbol{x}_t)$, sampling from DMs can be formulated by solving the diffusion ODE from $T$ to 0 Song et al. (2021b):

$$\frac{\mathrm{d}\boldsymbol{x}_t}{\mathrm{d}t} = f(t)\boldsymbol{x}_t + \frac{g^2(t)}{2\sigma_t}\epsilon_\theta(\boldsymbol{x}_t, t), \ \boldsymbol{x}_T \sim \mathcal{N}\left(\boldsymbol{0}, \hat{\sigma}^2\boldsymbol{I}\right). \tag{2.5}$$

With a different parameterization, the data prediction model $\boldsymbol{x}_\theta(\boldsymbol{x}_t, t)$ satisfies: $\boldsymbol{x}_\theta(\boldsymbol{x}_t, t) = (\boldsymbol{x}_t - \sigma_t \boldsymbol{\epsilon}_\theta(\boldsymbol{x}_t, t))/\alpha_t$ (Kingma et al., 2021). This results in an equivalent ODE-based diffusion process:

$$\frac{\mathrm{d}\boldsymbol{x}_t}{\mathrm{d}t} = \left(f(t) + \frac{g^2(t)}{2\sigma_t^2}\right)\boldsymbol{x}_t - \alpha_t \frac{g^2(t)}{2\sigma_t^2}\boldsymbol{x}_\theta(\boldsymbol{x}_t, t). \tag{2.6}$$

## 3 Conditional Entropy Reduction as a Catalyst for Denoising Diffusion

By applying the *variation-of-constants* formula (Hale & Lunel, 2013) to ODEs (2.5) and (2.6), then

$$\boldsymbol{x}_t = e^{\int_s^t f(r)\mathrm{d}r}\left(\int_s^t h_1(r)\boldsymbol{\epsilon}_\theta(\boldsymbol{x}_r, r)\,\mathrm{d}r + \boldsymbol{x}_s\right), \boldsymbol{x}_t = e^{h_2(t)}\left(-\int_s^t e^{-h_2(r)}\frac{\alpha_r g^2(r)}{2\sigma_r^2}\boldsymbol{x}_\theta(\boldsymbol{x}_r, r)\,\mathrm{d}r + \boldsymbol{x}_s\right), \tag{3.1}$$

where $h_1(r) := e^{-\int_s^r f(z)\mathrm{d}z}\frac{g^2(r)}{2\sigma_r}$, $h_2(r) := \int_s^r f(z) + \frac{g^2(z)}{2\sigma_z^2}\mathrm{d}z$, and $\boldsymbol{x}_s$ represents the given initial value. Subsequently, this two diffusion ODEs have a unified semi-linear solution formula.

**Remark 1** *Let the noise-prediction and data-prediction diffusion ODEs be defined by equations (2.5) and (2.6), respectively. A unified semi-linear solution formula for both ODEs is then given by:*

$$\boldsymbol{f}(\boldsymbol{x}_t) - \boldsymbol{f}(\boldsymbol{x}_s) = \int_{\kappa(s)}^{\kappa(t)} \boldsymbol{d}_\theta\left(\boldsymbol{x}_{\psi(\tau)}, \psi(\tau)\right)\mathrm{d}\tau, \tag{3.2}$$

*where $\psi(\kappa(t)) := t$, $\{\boldsymbol{f}(\boldsymbol{x}_t) := \boldsymbol{x}_t/\alpha_t, \kappa(t) := \sigma_t/\alpha_t\}$ when $\boldsymbol{d}_\theta$ represents the noise prediction model and $\{\boldsymbol{f}(\boldsymbol{x}_t) := \boldsymbol{x}_t/\sigma_t, \kappa(t) := \alpha_t/\sigma_t\}$ when $\boldsymbol{d}_\theta$ represents the data prediction model.*

For brevity, we refer to Eq. (3.2) as the *score-integral* process, as the denoiser $\boldsymbol{d}_\theta\left(\boldsymbol{x}_{\psi(\tau)}, \psi(\tau)\right)$ is often trained to approximate the score function. Note that the semi-linear nature of diffusion ODEs can potentially reduce the sampling error of DMs Lu et al. (2022a;b); Zhang & Chen (2023). Unless otherwise specified, the following discussion defaults to noise prediction models.

### 3.1 Denoising Iterations Formulated by Score-integral Estimation

Denote $h_{t_i} := \kappa(t_{i-1}) - \kappa(t_i)$, $\iota(\boldsymbol{x}_{t_{i-1}}) := \int_{\kappa(t_i)}^{\kappa(t_{i-1})} \boldsymbol{d}_\theta\left(\boldsymbol{x}_{\psi(\tau)}, \psi(\tau)\right)\mathrm{d}\tau$ and $\boldsymbol{d}_\theta^{(k)}\left(\boldsymbol{x}_{\psi(\tau)}, \psi(\tau)\right) := \frac{\mathrm{d}^k \boldsymbol{d}_\theta(\boldsymbol{x}_{\psi(\tau)}, \psi(\tau))}{\mathrm{d}\tau^k}$ as $k$-th order total derivative of $\boldsymbol{d}_\theta\left(\boldsymbol{x}_{\psi(\tau)}, \psi(\tau)\right)$ w.r.t. $\tau$. The Taylor expansion of $\boldsymbol{d}_\theta(\boldsymbol{x}_{t_{i-1}}, t_{i-1})$ at $\tau_{t_i}$ is

$$\boldsymbol{d}_\theta(\boldsymbol{x}_{t_{i-1}}, t_{i-1}) = \boldsymbol{d}_\theta(\boldsymbol{x}_{t_i}, t_i) + \sum_{k=1}^n \frac{h_{t_i}^k}{k!}\boldsymbol{d}_\theta^{(k)}(\boldsymbol{x}_{t_i}, t_i) + O(h_{t_i}^{n+1}). \tag{3.3}$$

Substituting this Taylor expansion into Eq. (3.2) to approximate $\iota(\boldsymbol{x}_{t_{i-1}})$ yields:

$$\tilde{\iota}(\boldsymbol{x}_{t_{i-1}}) = h_{t_i}\boldsymbol{d}_\theta(\boldsymbol{x}_{t_i}, t_i) + \sum_{k=1}^n \frac{h_{t_i}^{k+1}}{(k+1)!}\boldsymbol{d}_\theta^{(k)}(\boldsymbol{x}_{t_i}, t_i) + O(h_{t_i}^{n+2}). \tag{3.4}$$

Beyond the transformations within the solving space, this Taylor-based approximation establishes a generalized numerical iterative framework for solving the score-integral in DMs. For instance, when $n = 1$, the truncated Taylor approximation reduces to the well-known *DDIM* iterative algorithm Song et al. (2021a), as follows:

$$\boldsymbol{f}(\tilde{\boldsymbol{x}}_{t_{i-1}}) = \boldsymbol{f}(\tilde{\boldsymbol{x}}_{t_i}) + h_{t_i}\boldsymbol{d}_\theta(\tilde{\boldsymbol{x}}_{t_i}, t_i). \tag{3.5}$$

where $\tilde{\boldsymbol{x}}$ is obtained by the definition of $\boldsymbol{f}(\tilde{\boldsymbol{x}})$. Due to the lack of derivative information, higher-order algorithms can only be formulated by evaluating the derivatives. A widely used technique for evaluating derivatives is the finite difference (FD) method, which approximates $\boldsymbol{d}_\theta^{(k)}(\cdot, \cdot)$ as follows ($k \geq 1$):

$$\boldsymbol{d}_\theta^{(k)}(\boldsymbol{x}_t, t) = \frac{\boldsymbol{d}_\theta^{(k-1)}(\boldsymbol{x}_s, s) - \boldsymbol{d}_\theta^{(k-1)}(\boldsymbol{x}_t, t)}{\hat{h}_t} + O(\hat{h}_t). \tag{3.6}$$

Thus, a gradient estimation-based iteration can be obtained by truncating all higher-order derivatives:

$$\boldsymbol{f}(\tilde{\boldsymbol{x}}_{t_{i-1}}) = \boldsymbol{f}(\tilde{\boldsymbol{x}}_{t_i}) + h_{t_i}\boldsymbol{d}_\theta(\tilde{\boldsymbol{x}}_{t_i}, t_i) + \frac{h_{t_i}^2}{2}F_\theta(s_i, t_i), \tag{3.7}$$

where $F_\theta(s_i, t_i) := \frac{\boldsymbol{d}_\theta(\tilde{\boldsymbol{x}}_{s_i}, s_i) - \boldsymbol{d}_\theta(\tilde{\boldsymbol{x}}_{t_i}, t_i)}{\hat{h}_{t_i}}$, $\hat{h}_{t_i} := \kappa(s_i) - \kappa(t_i)$, $\hat{h}_{t_i} \neq 0$ and it is often satisfied that $\hat{h}_{t_i}/h_{t_i} \leq 1$.

### 3.2 CONDITIONAL ENTROPY REDUCTION IN DENOISING ITERATIONS

**Iterative Uncertainty Reduction: Theoretical Insights.** The semi-linear solution formula in Remark 1 provides a structured theoretical framework for analyzing the denoising diffusion process. By iteratively solving this formula, DMs refine noisy latent states closer to the data distribution. From an information-theoretic perspective, each iteration progressively reduces uncertainty from intermediate representations by leveraging the structured denoising mechanism. This uncertainty reduction can be formalized through the concept of *mutual information* between consecutive states Jaynes (1957):

$$I_p(\boldsymbol{x}_{t_i}; \boldsymbol{x}_{t_{i+1}}) = H_p(\boldsymbol{x}_{t_i}) - H_p(\boldsymbol{x}_{t_i}|\boldsymbol{x}_{t_{i+1}}), \tag{3.8}$$

where $H_p(\boldsymbol{x}_{t_i})$ is the entropy of state $\boldsymbol{x}_{t_i}$, and $H_p(\boldsymbol{x}_{t_i}|\boldsymbol{x}_{t_{i+1}})$ is the *conditional entropy* of $\boldsymbol{x}_{t_i}$ given $\boldsymbol{x}_{t_{i+1}}$. The conditional entropy $H_p(\boldsymbol{x}_{t_i}|\boldsymbol{x}_{t_{i+1}})$ quantifies the uncertainty in $\boldsymbol{x}_{t_i}$ after incorporating information from the subsequent state $\boldsymbol{x}_{t_{i+1}}$. A lower $H_p(\boldsymbol{x}_{t_i}|\boldsymbol{x}_{t_{i+1}})$ indicates that the method effectively utilizes information from $\boldsymbol{x}_{t_{i+1}}$ to refine the estimate of $\boldsymbol{x}_{t_i}$, driving the estimate of $\boldsymbol{x}_{t_i}$ closer to the target data distribution. Practically, this conditional entropy reduction aligns with the goal of minimizing reconstruction error during denoising, improving the quality of generated samples. This theoretical insight not only elucidates the uncertainty reduction mechanism but also provides an optimization criterion for improving the denoising process.

**Conditional Entropy in Gaussian Approximations.** In practical implementations of DMs Ho et al. (2020); Song et al. (2021b), the reverse transition distribution $p(\boldsymbol{x}_{t_i}|\boldsymbol{x}_{t_{i+1}}, \boldsymbol{x}_0)$ is commonly approximated as a Gaussian distribution under the *Markov assumption*. For brevity, $p(\boldsymbol{x}_{t_i}|\boldsymbol{x}_{t_{i+1}}, \boldsymbol{x}_0)$ is often abbreviated as $p(\boldsymbol{x}_{t_i}|\boldsymbol{x}_{t_{i+1}})$. Then, this reverse transition distribution can be expressed as

$$p(\boldsymbol{x}_{t_i}|\boldsymbol{x}_{t_{i+1}}) := p(\boldsymbol{x}_{t_i}|\boldsymbol{x}_{t_{i+1}}, \boldsymbol{x}_0) \approx \mathcal{N}(\boldsymbol{\mu}_{t_i}, \boldsymbol{\Sigma}_{t_i}), \tag{3.9}$$

where $\boldsymbol{\mu}_{t_i}$ and $\boldsymbol{\Sigma}_{t_i}$ are derived using Bayes' rule from the forward diffusion process. This Gaussian approximation is widely used for simplifying model training and theoretical analysis, despite potential deviations at extreme steps, as noted in prior works Song et al. (2021b); Luo (2022); Bao et al. (2022); Karras et al. (2022). Under this approximation, the conditional entropy $H_p(\boldsymbol{x}_{t_i}|\boldsymbol{x}_{t_{i+1}})$ simplifies to

$$H_p(\boldsymbol{x}_{t_i}|\boldsymbol{x}_{t_{i+1}}) \approx \frac{d}{2}(\log 2\pi + 1) + \frac{1}{2} \log |\text{Var}(\boldsymbol{x}_{t_i}|\boldsymbol{x}_{t_{i+1}})|, \tag{3.10}$$

where $d$ is the dimensionality of $\boldsymbol{x}$, and $\text{Var}(\boldsymbol{x}_{t_i}|\boldsymbol{x}_{t_{i+1}})$ is the conditional variance. This expression provides a tractable framework for analyzing conditional entropy reduction during the denoising iteration, as it establishes a direct relationship between conditional entropy and the variance. Note that the conditional entropy $H_p(\boldsymbol{x}_{t_i}|\boldsymbol{x}_{t_{i+1}})$ is intrinsically tied to the conditional variance $\text{Var}(\boldsymbol{x}_{t_i}|\boldsymbol{x}_{t_{i+1}})$:

$$H_p(\boldsymbol{x}_{t_i}|\boldsymbol{x}_{t_{i+1}}) \propto \log |\text{Var}(\boldsymbol{x}_{t_i}|\boldsymbol{x}_{t_{i+1}})|. \tag{3.11}$$

Thus, Eq. (3.11) suggests that minimizing variance directly optimizes conditional entropy reduction.
**Variance-Driven Conditional Entropy Reduction in Gradient-Based Iterations.** Building on the established relationship between conditional variance and entropy, we derive several analytical results that provide insights into the conditional entropy reduction achieved by gradient-based denoising iterations. For instance, under suitable conditions, our analysis suggests that gradient estimation-based iterations (Eq. (3.7)) can effectively drive significant reductions in conditional entropy.

To simplify the analysis, we assume that the estimated noise $\boldsymbol{\epsilon}_\theta(\cdot)$ at different timesteps is independent. While the forward process has the Markov property, our assumption mainly stems from practical considerations in training. Specifically, the training objective of Eq. (2.4) in DDPMs Ho et al. (2020) minimizes the mean squared error at each timestep independently, which aligns with this assumption. Although adopting a parameter-sharing setting across timesteps in the noise prediction model may involve a compromise on the assumption of independence, prior works Song et al. (2021a) indicate that these dependencies have minimal impact on model performance. This makes the independence assumption Ho et al. (2020) a reasonable and practical surrogate for theoretical analysis.
Under this independence assumption, we derive the Proposition 3.1, with the proof in Appendix B.1.

**Proposition 3.1** *The gradient-based denoising iteration in Eq. (3.7) tends to reduce conditional entropy more efficiently than the first-order iteration in Eq. (3.5) when* $\frac{h_{t_i}}{\hat{h}_{t_i}} \in \left[1, \frac{4 \cdot \text{Var}(\boldsymbol{\epsilon}_\theta(\tilde{\boldsymbol{x}}_{t_i}, t_i))}{\text{Var}(\boldsymbol{\epsilon}_\theta(\tilde{\boldsymbol{x}}_{s_i}, s_i)) + \text{Var}(\boldsymbol{\epsilon}_\theta(\tilde{\boldsymbol{x}}_{t_i}, t_i))}\right]$.

Intuitively, this result reveals that gradient-based denoising iterations can achieve greater reductions in uncertainty compared to first-order methods when the step-size ratio is properly chosen. As the reverse process in DMs aims to estimate $p(\boldsymbol{x}_t|\boldsymbol{x}_{t+1}, \boldsymbol{x}_0)$ Ho et al. (2020); Luo (2022), we examine $\text{Var}(\boldsymbol{\epsilon}_\theta(\tilde{\boldsymbol{x}}_t, t) \mid \boldsymbol{x}_0)$ to capture the model's uncertainty in noise prediction conditioned on the clean data. For brevity, we denote this variance as $\text{Var}(\boldsymbol{d}_\theta(\tilde{\boldsymbol{x}}_t, t))$ throughout the paper. Based on this consideration, we can establish the practical interval for Proposition 3.1 using the prior-like conditional variance.

**Proposition 3.2** *In the forward process of DMs, the clean data at each step can be expressed by $x_0 = x_t/\alpha_t - \sigma_t/\alpha_t \epsilon$. If we assume that $\text{Var}(\epsilon_\theta(\tilde{x}_t, t)) \propto \sigma_t^2/\alpha_t^2$ to quantify the extent of deviation from the clean data. Under this prior-like assumption, we obtain $\frac{\text{Var}(\epsilon_\theta(\tilde{x}_{s_i}, s_i))}{\text{Var}(\epsilon_\theta(\tilde{x}_{t_i}, t_i))} = \frac{\text{SNR}(t_i)}{\text{SNR}(s_i)}$. Then, the relative condition of conditional entropy reduction in Propostion 3.1 is $h_{t_i}/\hat{h}_{t_i} \in \left[1, \frac{4 \text{ SNR}(s_i)}{\text{SNR}(t_i) + \text{SNR}(s_i)}\right]$.*

Additionally, interpreting the denoising numerical iterative mechanisms through the lens of conditional entropy reduction offers deeper insights into accelerated denoising diffusion solvers, such as the widely recognized accelerated iterations in DPM-Solver Lu et al. (2022a) and EDM Karras et al. (2022). Building on this insight, we present the following proposition, with details in Appendix B.2.

**Proposition 3.3** *The exponential integrator-based iterations in DPM-Solver and the Heun iterations in EDM can be interpreted as specific instances of accelerated denoising mechanisms driven by conditional entropy reduction, thereby distinguishing them from conventional gradient-based methods.*

Finally, based on our comprehensive analysis of the differences in conditional entropy reduction between denoising iterations using data-prediction and noise-prediction parameterization, we derive the following conclusion. The detailed proof is provided in Appendix B.4.

**Proposition 3.4** *Assuming that the injected noise at different time steps in DM is mutually independent, denoising iterations using data-prediction parameterization are more effective at reducing conditional entropy than those using noise-prediction parameterization in a well-trained DM.*

Proposition 3.4 highlights the key advantage of data-prediction:it directly aligns with the target distribution $x_0$, bypassing the intermediate noise-to-data mapping $\epsilon_t \mapsto x_t \mapsto x_0$, which can accumulate errors, especially in late timesteps with high noise variance (or few-step sampling). By minimizing conditional entropy more effectively, data-prediction reduces uncertainty in $x_0$ without relying on intermediate transformations. Nonetheless, this advantage is contingent on the training quality. If the model struggles to accurately predict $x_0$, noise-prediction parameterization, which treats timesteps more uniformly, may perform better in practice.

In summary, the perspective of conditional entropy reduction deepens our understanding of denoising mechanisms in diffusion model sampling, while the variance-driven approach provides valuable insights into the design of efficient denoising algorithms.

# 4 VARIANCE-DRIVEN EFFICIENT CONDITIONAL ENTROPY REDUCTION ITERATION

In this section, we elucidate the approach for improving both single-step and multi-step numerical iterations through conditional entropy reduction. Building on prior-like model variance assumptions, we derive several efficient iteration rules for conditional entropy reduction and establish the convergence orders of these iterations. Finally, we further optimize these iteration rules by refining the conditional variance with the actual state differences observed during the iterative process.

## 4.1 SINGLE-STEP ITERATION WITH CONDITIONAL ENTROPY REDUCTION

*One key insight is that* the model parameter $\epsilon_\theta(\tilde{x}_{s_i}, s_i)$ can be further leveraged to enhance gradient estimation-based iteration, as observed in Eq. (3.7) and supported by conditional entropy analysis, without additional model parameters. Formally, the improvement iteration can be defined as follows:

$$f(\tilde{x}_{t_{i-1}}) = f(\tilde{x}_{t_i}) + h_{t_i}\left(\gamma_i d_\theta(\tilde{x}_{s_i}, s_i) + (1 - \gamma_i)d_\theta(\tilde{x}_{t_i}, t_i)\right) + \frac{h_{t_i}^2}{2}F_\theta(s_i, t_i), \quad (4.1)$$

where $\gamma_i \in (0, 1]$. This improved iteration shares the same limit state as the vanilla gradient estimation-based denoising iteration in Eq. (3.7) when $s_i \to t_i$. For convenience, we refer to the standard gradient estimation-based iteration as the **FD-based** iteration. In the analysis of conditional entropy, we can compare the different components of Eq. (3.7) and Eq. (4.1). Therefore, the variance of the key distinct components in each conditional distribution is as follows:

$$\text{Var}_{p_1} = h_{t_i}^2 \cdot \text{Var}(\epsilon_\theta(\tilde{x}_{t_i}, t_i)), \quad \text{Var}_{p_2}(\gamma_i) = h_{t_i}^2\left(\gamma_i^2 \cdot \text{Var}(\epsilon_\theta(\tilde{x}_{s_i}, s_i)) + (1 - \gamma_i)^2 \cdot \text{Var}(\epsilon_\theta(\tilde{x}_{t_i}, t_i))\right). \quad (4.2)$$

Then, the difference in conditional entropy between the two gradient estimation-based iterations is

$$\Delta H(p) = \frac{1}{2}\log\frac{\text{Var}_{p_2}(\gamma_i)}{\text{Var}_{p_1}} = \frac{1}{2}\log\left(1 - 2\gamma_i + \gamma_i^2 + \gamma_i^2\frac{\text{Var}(\epsilon_\theta(\tilde{x}_{s_i}, s_i))}{\text{Var}(\epsilon_\theta(\tilde{x}_{t_i}, t_i))}\right). \quad (4.3)$$

Due to $\gamma_i \in (0, 1]$ and $\text{SNR}(t_i) \leq \text{SNR}(s_i)$, $\Delta H(p) \leq 0$ consistently holds under the assumption that $\text{Var}(\epsilon_\theta(\tilde{x}_t, t)) \propto \sigma_t^2/\alpha_t^2$. Therefore, this improved iteration can more efficiently reduce conditional entropy compared to the vanilla iteration by using subsequent model parameters in lower-variance regions as guidance. Consequently, based on $\Delta H(p) \leq 0$, we have the following proposition.

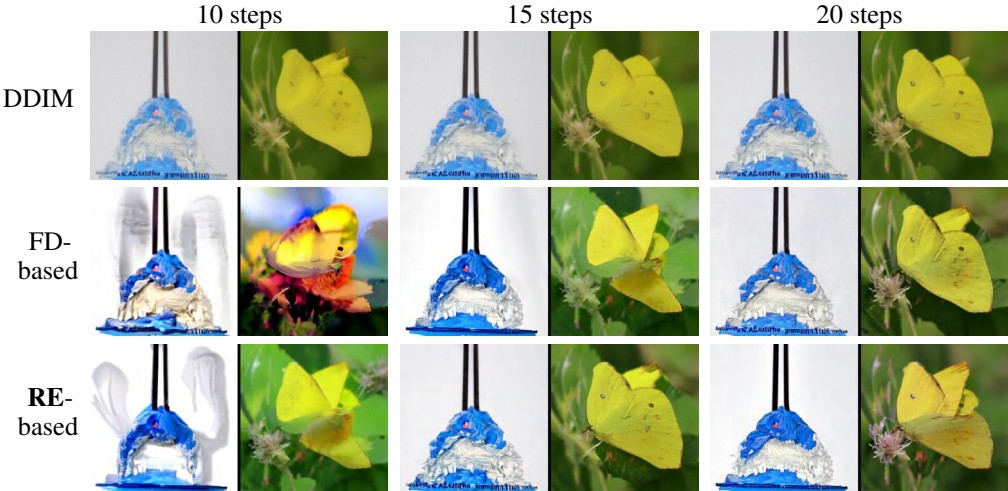

(a) Single-step Iteration on CIFAR-10 (Discrete)  (b) Single-step Iteration on CelebA-64 (Discrete)

Figure 2: Comparisons of FID ↓ computed by **RE-based** and FD-based iterations demonstrate that efficient entropy reduction consistently enhances image quality across various ablation scenarios.

Figure 3: Samples were generated from a pre-trained DM on the ImageNet 256×256 dataset using noise-prediction parameterization with 10-20 single-step iterations. The sample results indicate that **RE-based** iterations can improve sample quality by reducing the conditional variance.

**Proposition 4.1** *The iteration specified in Eq. (4.1) consistently achieves a more efficient reduction in conditional entropy than the FD-based iteration. Then, an efficient improvement interval for $\gamma_i$ is recommended as $\left[\frac{\text{SNR}(t_i)}{\text{SNR}(t_i)+\text{SNR}(s_i)}, \frac{\max\{2\cdot\text{SNR}(t_i),\ \text{SNR}(s_i)\}}{\text{SNR}(t_i)+\text{SNR}(s_i)}\right]$. For clarity, we identify this iteration that enhances denoising efficiency by **r**educing conditional **e**ntropy as the **RE-based** iteration.*

Accordingly, Proposition 4.1 demonstrates that the RE-based iteration can consistently surpass the FD-based iteration in reducing conditional entropy. In the following, we provide the convergence guarantees in Theorem 4.1 for the RE-based iteration, the proof is provided in Appendix C.2.

**Theorem 4.1** *If $d_\theta(x_t, t)$ satisfies Assumption C.1, the RE-based iteration constitutes a globally convergent second-order iterative algorithm.*

Consequently, although the RE-based iteration in Eq. (4.1) shares the same order of convergence as the FD-based iteration, the primary distinction between the RE-based and FD-based iterations lies in their handling of conditional variance, which improves the denoising diffusion process by enabling more efficient conditional entropy reduction with the same model parameters.

### 4.2 Multi-step Iteration with Conditional Entropy Reduction

This section focuses on the multi-step iteration with a step size determined by the two adjacent time points. Our discussion focuses on the conditional entropy reduction in multi-step iterations that leverage *data-prediction parameterization*, as this approach has demonstrated its superiority through our theoretical result presented in Proposition 3.4 and the empirical evidence from the earlier study in Lu et al. (2022b). The difference analysis of multi-step iterations is provided in Appendix B.3. Formally, the iteration with a step size determined by two adjoint time points can be written as:

$$f(\tilde{x}_{t_{i-1}}) = f(\tilde{x}_{t_i}) + h_{t_i} d_\theta(\tilde{x}_{t_i}, t_i) + \frac{h_{t_i}^2}{2} B_\theta(t_i, t_{i+1}), \tag{4.4}$$

where $B_\theta(t_i, t_{i+1}) := \frac{d_\theta(\tilde{x}_{t_i}, t_i) - d_\theta(\tilde{x}_{t_{i+1}}, t_{i+1})}{h_{t_{i+1}}}$. In this iteration, the step size $|h_{t_i}|$ is smaller than the step size $|h_{t_i} - h_{t_{i+1}}|$ used in gradient estimation-based iterations for the single-step case. As smaller step sizes reduce conditional entropy according to Eq. (3.10), the iteration in Eq. (4.4) offers greater potential for improving the denoising process compared to single-step counterparts.

A straightforward improvement of the iteration in Eq. (4.4) can be formulated as follows:

$$f(\tilde{x}_{t_{i-1}}) = f(\tilde{x}_{t_i}) + h_{t_i} d_\theta(\tilde{x}_{t_i}, t_i) + \frac{h_{t_i}^2}{2} B_\theta(t_i, l_i), \tag{4.5}$$

where $d_\theta(\tilde{x}_{l_i}, l_i) = \zeta_i d_\theta(\tilde{x}_{t_i}, t_i) + (1 - \zeta_i) d_\theta(\tilde{x}_{t_{i+1}}, t_{i+1})$ represents a linear interpolation of the model parameters between times $t_i$ and $t_{i+1}$. Similarly, the implicit improvement approach is as follows:

$$f(\tilde{x}_{t_{i-1}}) = f(\tilde{x}_{t_i}) + h_{t_i} d_\theta(\tilde{x}_{t_i}, t_i) + \frac{h_{t_i}^2}{2} B_\theta(s_i, t_i), \tag{4.6}$$

where $d_\theta(\tilde{x}_{s_i}, s_i) = \zeta_i d_\theta(\tilde{x}_{t_{i-1}}, t_{i-1}) + (1 - \zeta_i) d_\theta(\tilde{x}_{t_i}, t_i)$. Note that both iterations can be unified as

$$f(\tilde{x}_{t_{i-1}}) = f(\tilde{x}_{t_i}) + h_{t_i} d_\theta(\tilde{x}_{t_i}, t_i) + \frac{h_{t_i}^2}{2} \zeta_i \bar{B}_\theta(t_i; u_i), \tag{4.7}$$

where $\bar{B}_\theta(t_i; u_i) = B_\theta(s_i, t_i)$ when $u_i = s_i$, and $\bar{B}_\theta(t_i; u_i) = B_\theta(t_i, l_i)$ when $u_i = l_i$. Similar to the case of single-step iterations, the following conditional entropy relation also holds for multi-step iterations.

**Remark 2** *The improved multi-step iterations in Eq. (4.7) reduce the conditional entropy of the vanilla multi-step iterations in Eq. (4.4) by leveraging model parameters from low-variance regions.*

However, a key question arises: how should $\zeta_i$ and $\hat{h}_{t_i}$ be determined? In the data prediction model, $d_\theta(\tilde{x}_{t_i}, t_i)$ is designed to directly predict the clean data $x_0$ from the intermediate noisy data $\tilde{x}_{t_i}$. Since $\tilde{x}_{t_i}$ is perturbed by Gaussian noise with a standard deviation $\sigma_{t_i}$, $\sigma_{t_i}$ reflects the amount of noise present at time step $t_i$. Then, for the interpolation of $d_\theta(\tilde{x}_{s_i}, s_i)$, we have the following proposition.

**Proposition 4.2** *If assume that* $\mathrm{Var}(d_\theta(\tilde{x}_{t_i}, t_i)) \propto \sigma_{t_i}^2$, *the minimizing variance can be achieved when* $\zeta_i = \frac{\sigma_{t_{i-1}}^2}{\sigma_{t_i}^2 + \sigma_{t_{i-1}}^2}$ *for* $d_\theta(\tilde{x}_{s_i}, s_i)$. *For* $d_\theta(\tilde{x}_{l_i}, l_i)$, *the optimal choice of lower variance is* $\zeta_i = \frac{\sigma_{t_i}^2}{\sigma_{t_i}^2 + \sigma_{t_{i+1}}^2}$.

*One key insight is that* we can further improve the denoising iteration in Eq. (4.4) with gradient estimation by incorporating $B_\theta(t_i, s_i)$ and $B_\theta(s_i, t_i)$ as follows:

$$f(\tilde{x}_{t_{i-1}}) = f(\tilde{x}_{t_i}) + h_{t_i} d_\theta(\tilde{x}_{t_i}, t_i) + \frac{h_{t_i}^2}{2} (\eta_i B_\theta(s_i, t_i) + (1 - \eta_i) B_\theta(t_i, l_i)). \tag{4.8}$$

In Eq. (4.8), $\eta_i$ determines the variance of the gradient term. From the perspective of conditional entropy reduction, we can reduce this variance by establishing an optimization objective that measures the differences between the corresponding states. Thus, in the next section, we will discuss the optimized $\eta_i$ and $\zeta_i$ based on the actual state differences observed during the iterative process.

### 4.3 Improving RE-based Iterations with Actual State Differences

In the previous sections, we derived the RE-based numerical iteration to reduce conditional entropy, grounded in the model's prior-like variance. In this section, the RE-based iteration is further optimized by refining the model variance with the actual state differences observed during the iterative process. **Improving $\zeta_i$ with Evolution State Differences.** Our goal is to refine $\zeta_i$ in the RE-based iteration. What follows is the optimized $\zeta_i$ by formulating an optimization objective. Denote $G(\zeta_i) := \zeta_i d_\theta(\tilde{x}_{s_i}, s_i) + (1 - \zeta_i) d_\theta(\tilde{x}_{t_i}, t_i)$, where $\zeta_i \in (0, 1]$. On one hand, we can rewrite the RE-based iteration as

$$f(\tilde{x}_{t_i}) = f(\tilde{x}_{t_{i-1}}) - h_{t_i} G(\zeta_i) - \frac{h_{t_i}^2}{2} F_\theta(s_i, t_i). \tag{4.9}$$

Notice that $\tilde{x}_{t_i}$ in Eq. (4.9) is determined by $\zeta_i$. On the other hand, we can consider $\tilde{x}_{t_{i-1}}$ as a starting point and perform a inverse iterative from $t_{i-1}$ to $t_i$ to approximate $\tilde{x}_{t_i}$. The inverse iterative formula is

$$f(x_s) - f(x_t) = \int_{\kappa(t)}^{\kappa(s)} d_\theta(x_{\psi(\tau)}, \psi(\tau)) d\tau. \tag{4.10}$$

Similar to the score-integral iteration in Eq. (3.7), this inverse integral can be estimated by

$$\tilde{\Delta}_{t_i}^{\mathrm{reverse}} = -h_{t_i} d_\theta(\tilde{x}_{t_{i-1}}, t_{i-1}) + \frac{h_{t_i}^2}{2} F_\theta(s_i, t_{i-1}). \tag{4.11}$$

Based on equations (4.10) and (4.11), we obtain a new estimation $\hat{x}_{t_i}$ for the state $x_{t_i}$ as follows:

$$f(\hat{x}_{t_i}) = f(\tilde{x}_{t_{i-1}}) - h_{t_i} d_\theta(\tilde{x}_{t_{i-1}}, t_{i-1}) + \frac{h_{t_i}^2}{2} F_\theta(s_i, t_{i-1}). \tag{4.12}$$

Drawing inspiration from equations (4.9) and (4.12), we can determine $\zeta_i$ by minimizing the differences between two estimations. Then, the optimization objective for $\zeta_i$ is defined as follows:

$$\min_{\zeta_i \in (0,1]} \mathcal{L}_1(\zeta_i) := \left\| (\tilde{x}_{t_i} - x_{t_i}) + (\hat{x}_{t_i} - x_{t_i}) \right\|_F, \tag{4.13}$$

Directly solving this objective is challenging, as $x_{t_i}$ is unknown in practice. Fortunately, there exists an tractable error upper bound (*EUB*) for $\mathcal{L}_1(\zeta_i)$. Specifically, denote $\mathcal{L}_{1s}(\zeta_i) := \|\tilde{x}_{t_i} + \hat{x}_{t_i}\|_F$, we have

$$\mathcal{L}_1(\zeta_i) = \left\| \tilde{x}_{t_i} + \hat{x}_{t_i} - 2x_{t_i} \right\|_F \leq \left\| \tilde{x}_{t_i} + \hat{x}_{t_i} \right\|_F + \left\| 2x_{t_i} \right\|_F = \mathcal{L}_{1s}(\zeta_i) + \left\| 2x_{t_i} \right\|_F, \tag{4.14}$$

where $\|2x_{t_i}\|_F$ can be viewed as a specific regularization term. Since $\|2x_{t_i}\|_F$ is independent of the target $\zeta_i$, we can optimize the vanilla $\mathcal{L}_1(\zeta_i)$ by minimizing $\mathcal{L}_{1s}(\zeta_i)$ according to Eq. (4.14). Then, the optimized $\zeta_i$ can be obtained by solving $\min \mathcal{L}_{1s}(\zeta_i)$ with a small regularization using $\tilde{x}_{t_i}$. For example, denote $P(\tilde{x}_{t_{i-1}}^p) := \hat{x}_{t_i} + \frac{\sigma_{t_i}}{\sigma_{t_{i-1}}} \tilde{x}_{t_{i-1}}^p - \sigma_{t_i} \frac{h_{t_i}^2}{2} F_\theta(s_i, t_i)$, where $\tilde{x}_{t_{i-1}}^p$ can be obtained by prior RE-based iteration. Then, the simplified optimization objective $\mathcal{L}_{1s}(\zeta_i)$ can be rewritten as:

$$\mathcal{L}_{1s}(\zeta_i) = \left\| P(\tilde{x}_{t_{i-1}}^p) - \sigma_{t_i} h_{t_i} G(\zeta_i) \right\|_F. \tag{4.15}$$

**Practical Considerations.** In practice, the constraints on $\zeta_i$ can hinder its computational efficiency. To address this, we adopt an optimization-guided streamlined approach for determining $\zeta_i$. Specifically, we observe that the optimization objective admits a closed-form solution, as presented in Lemma 4.1, when the constraints on $\zeta_i$ are relaxed. These constraints are then satisfied by applying an activation function to map the closed-form solutions. This optimization-driven streamlined approach not only captures the actual differences in states during the iterative process, but also circumvents the computational cost of solving constrained optimization problems iteratively Boyd et al. (2011).

---

**Algorithm 1** Denoising Diffusion Sampling by Variance-Driven Conditional Entropy Reduction.

---

**Require:** initial value $x_T$, time schedule $\{t_i\}_{i=0}^N$, model $d_\theta$.
1: $\tilde{x}_{t_N} \leftarrow x_T$, $h_{t_i} \leftarrow \kappa(t_{i-1}) - \kappa(t_i)$
2: **for** $i \leftarrow N$ to $1$ **do**
3:      $f(\tilde{x}_{t_i}) \leftarrow f(\tilde{x}_{t_{i+1}}) + h_{t_{i+1}} d_\theta(\tilde{x}_{t_{i+1}}, t_{i+1})$
4:      $\zeta_i = r_i h_{t_i}$, where $r_i$ is used to balance the prior-like variance, such as the log-SNR ratio.
5:      $f(\tilde{x}_{t_{i-1}}) = f(\tilde{x}_{t_i}) + h_{t_i} d_\theta(\tilde{x}_{t_i}, t_i) + \frac{h_{t_i}^2}{2} B_\theta(t_i, l_i)$
6:      $\eta_i = \text{Sigmoid}\left(|\eta_i^*|\right)$, where $\eta_i^*$ is computed using Eq. (4.19).
7:      $B_\theta(t_i) \leftarrow \frac{\eta_i}{2} B_\theta(s_i, t_i) + \left(1 - \frac{\eta_i}{2}\right) B_\theta(t_i, l_i)$
8:      $\zeta_i = \text{Sigmoid}\left(|\zeta_i^*| - \mu\right)$, where $\mu$ is the shift parameter, and $\zeta_i^*$ is computed using Eq. (4.16).
9:      $f(\tilde{x}_{t_{i-1}}) \leftarrow f(\tilde{x}_{t_i}) + h_{t_i} d_\theta(\tilde{x}_{t_i}, t_i) + \frac{h_{t_i}^2}{2} \zeta_i B_\theta(t_i)$
10: **end for**
     **return** : $\tilde{x}_0$.

---

**Lemma 4.1** *The minimizing problem* $\min_{\zeta_i} \mathcal{L}_{1s}^2(\zeta_i)$ *possesses the following closed-form solution:*

$$\zeta_i^* = -\frac{vec^T(D_i) vec(\tilde{P}_i)}{\sigma_{t_i} h_{t_i} vec^T(D_i) vec(D_i)}, \tag{4.16}$$

*where* $\tilde{P}_i := P(\tilde{x}_{t_{i-1}}^p) - \sigma_{t_i} h_{t_i} x_\theta(\tilde{x}_{t_i}, t_i)$, $D_i := x_\theta(\tilde{x}_{s_i}, s_i) - x_\theta(\tilde{x}_{t_i}, t_i)$, *and* $vec(\cdot)$ *denotes the vectorization operation. The proof details can be found in Appendix C.3.*

**Improving $\eta_i$ with Balanced Difference Techniques.** Our goal is to refine the RE-based iteration by optimize $\eta_i$ with the available information *at current step*. Denote $\tilde{\Delta}_{t_i}^g = \eta_i F_\theta(t_{i-1}, t_i) + (1 - \eta_i) F_\theta(t_{i+1}, t_i)$. We can define the estimation error of derivative at point $\tau_{t_i}$ as $E(t_{i-1}, t_i) := F_\theta(t_{i-1}, t_i) - d_\theta^{(1)}(x_{t_i}, t_i)$. For balancing the estimation errors, we formulate the following optimization objective:

$$\min_{\eta_i \in (0,1]} \mathcal{L}_2(\eta_i) := \|\eta_i E(t_{i-1}, t_i) + (1 - \eta_i) E(t_{i+1}, t_i)\|_F. \tag{4.17}$$

We can rewrite $\mathcal{L}_2(\eta_i)$ as $\mathcal{L}_2(\eta_i) = \left\| \tilde{\Delta}_{t_i}^g - d_\theta^{(1)}(x_{t_i}, t_i) \right\|_F$. Denote $\mathcal{L}_{2s}(\eta_i) := \|\tilde{\Delta}_{t_i}^g\|_F$. Then, we have

$$\mathcal{L}_2(\eta_i) = \left\| \tilde{\Delta}_{t_i}^g - \epsilon_\theta^{(1)}(x_{t_i}, t_i) \right\|_F \leq \mathcal{L}_{2s}(\eta_i) + \left\| d_\theta^{(1)}(x_{t_i}, t_i) \right\|_F, \tag{4.18}$$

where $d_\theta^{(1)}(x_{t_i}, t_i)$ can be regarded as a specific regularization term independent of the target $\eta_i$. The optimized $\eta_i$ can be obtained by minimizing the tractable EUB term $\mathcal{L}_{2s}(\eta_i)$. Similarly, for practical considerations, we employ optimization-guided streamlined approach for determining $\eta_i$. We first calculate the closed-form solution outlined in Lemma 4.2. The refined $\eta_i$ is then obtained by mapping these solutions into the constrained space using an activation function, such as the Sigmoid function.

**Lemma 4.2** *The minimizing problem* $\min\limits_{\eta_i} \mathcal{L}_{2s}^2(\eta_i)$ *possesses the following closed-form solution:*

$$\eta_i^* = -\frac{vec^T(\tilde{F}_i)vec(F_\theta(t_{i+1}, t_i))}{vec^T(\tilde{F}_i)vec(\tilde{F}_i)}, \tag{4.19}$$

*where* $\tilde{F}_i := F_\theta(t_{i+1}, t_i) - F_\theta(t_{i-1}, t_i)$. *The proof process is similar to that of Lemma 4.1.*

Consequently, by integrating the optimized $\zeta_i$ and $\eta_i$ into the iterations of Eq. (4.8), we obtain the refined RE iterations. Algorithm 1 outlines this improved iteration process, which exhibits second-order global convergence, and the proof details are provided in Appendix C.4.

## 5 EXPERIMENTS

In this section, we experimentally validate our approach in both single-step and multi-step scenarios, demonstrating that variance-driven conditional entropy reduction improves the denoising process of pre-trained diffusion model in both pixel and latent spaces. This method effectively extends the capabilities of existing training-free ODE samplers without incurring additional computational overhead. We compare Algorithm 1 against the baseline methods on Stable Diffusion, as illustrated in Figure 7. More implementation details and additional results are provided in Appendix D.

### 5.1 SINGLE-STEP ITERATIONS

In the single-step iterations, we adopt DPM-Solver Lu et al. (2022a) as our baseline, focusing on denoising iterations based on noise prediction parameterization. Each step of the single-step mechanism only requires information from the starting point and prior to the endpoint. To ensure variance reduction in each step, we configure the step size ratio $r_i$ following the effective interval defined in Proposition 4.1 and $\gamma_i$ as specified in Proposition 3.2. As a specific instance of RE-based iterations (Proposition 3.3), DPM-Solver has demonstrated its effectiveness over traditional gradient-based iterations. We further validate RE-based iterations through experiments on CIFAR-10 Krizhevsky (2009), CelebA 64 Liu et al. (2015), and ImageNet 256 Deng et al. (2009), comparing them against solvers such as DDPM Ho et al. (2020), DDIM Song et al. (2021a), and Analytic-DDPM Bao et al. (2022). Results (Figures 2, 3, and 1) consistently show improved performance due to improved variance reduction. On CIFAR-10, the RE-based iteration achieves a *3.15* FID with only *84* NFEs, surpassing DDPM's Ho et al. (2020) 3.17 FID with 1000 NFEs, improving quality while realizing approximately 10× acceleration. Additional comparisons are provided in Figure 4.

### 5.2 MULTI-STEP ITERATIONS

In the multi-step iterations, we primarily adopt DPM-Solver++ Lu et al. (2022b) as our baseline, focusing on denoising iterations based on data-prediction parameterization. Leveraging the multi-step mechanism enables us to utilize marginally more model information compared to single-step approaches. This allows us to optimize conditional variance through actual state differences, thereby circumventing the limitations imposed by prior-like variance assumptions. To demonstrate the effectiveness of efficient conditional entropy reduction in improving the denoising process of DMs, we propose Algorithm 1, which improves denoising diffusion sampling by leveraging the variance-driven approach aimed at minimizing actual state differences. Notably, DPM-Solver-v3 Zheng et al. (2023b) recently introduced a novel optimization-based parameterization scheme, distinct from data-prediction and noise-prediction parameterizations, achieving impressive sampling performance, particularly on CIFAR-10. Therefore, we adopt DPM-Solver-v3 as our baseline method for CIFAR-10 experiments, considering its demonstrated advantages in optimized parameterization on this dataset. We evaluated the RE-based iterations against widely-recognized benchmark solvers, including DPM-Solver++ Lu et al. (2022b), DEIS Zhang & Chen (2023), UniPC Zhao et al. (2024), and DPM-Solver-v3 Zheng et al. (2023a) on both CIFAR-10 and ImageNet 256 datasets. The experimental

results (Tables 3, 2) demonstrate that the variance-driven conditional entropy reduction consistently improves sampling performance. Furthermore, we validated the effectiveness of our approach on pre-trained models in the latent space, such as Stable Diffusion, with results illustrated in Figure 7.

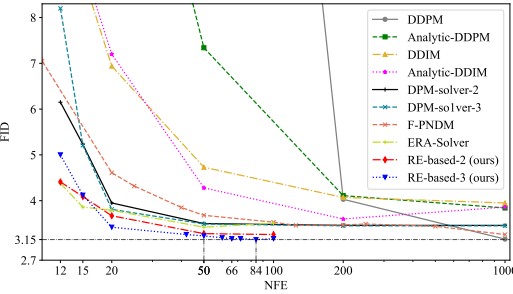

Figure 4: Comparisons of FID ↓ for single-step RE iterations on discrete DMs in CIFAR-10.

| Discrete | Continuous | Cond. EDM |
|---|---|---|
| 3.17 | 2.55 | 1.79 |
| DDPM | Hybrid PC | EDM |
| 3.26 | 2.64 | 1.79 |
| F-PNDM | DPM-Solver-v3 | Heun's 2nd |
| **3.15** | **2.41** | **1.76** |
| RE-based | RE-based | RE-based |

Table 1: The comparison for the performance limits of sampling methods on CIFAR-10 Krizhevsky (2009) indicates that RE-based iterations can further improve the denoising process.

Table 2: Quantitative results of the gradient estimation-based denoising iterations on ImageNet-256 Deng et al. (2009). We report the FID ↓ evaluated on 10k samples for various NFEs.

| Method | Model | NFE | | | | | | |
|---|---|---|---|---|---|---|---|---|
| | | 5 | 6 | 8 | 10 | 12 | 15 | 20 |
| DPM-Solver++ | | 15.69 | 11.65 | 9.06 | 8.29 | 7.94 | 7.70 | 7.48 |
| UniPC | Guided-Diffusion | 15.03 | 11.30 | 9.07 | 8.36 | 8.01 | 7.71 | 7.47 |
| DPM-Solver-v3 | ($s = 2$) | 14.92 | 11.13 | 8.98 | 8.14 | 7.93 | 7.70 | 7.42 |
| RE-based | | **13.98** | **10.98** | **8.84** | **8.14** | **7.79** | **7.48** | **7.25** |

Table 3: Quantitative results of the gradient estimation-based denoising iterations on CIFAR10. We report the FID ↓ evaluated on 50k samples for the different NFEs. We directly borrow the results of reported in the original paper of other methods.

| Method | Model | NFE | | | | | | |
|---|---|---|---|---|---|---|---|---|
| | | 5 | 6 | 8 | 10 | 12 | 15 | 20 |
| DEIS | | 15.37 | \ | \ | 4.17 | \ | 3.37 | 2.86 |
| DPM-Solver++ | | 28.53 | 13.48 | 5.34 | 4.01 | 4.04 | 3.32 | 2.90 |
| UniPC | ScoreSDE | 23.71 | 10.41 | 5.16 | 3.93 | 3.88 | 3.05 | 2.73 |
| DPM-Solver-v3 | | **12.76** | **7.40** | **3.94** | 3.40 | 3.24 | 2.91 | 2.71 |
| RE-based | | 13.54 | 8.56 | 4.11 | **3.38** | **3.22** | **2.76** | **2.42** |
| Heun's 2nd | | 320.80 | 103.86 | 39.66 | 16.57 | 7.59 | 4.76 | 2.51 |
| DPM-Solver++ | | 24.54 | 11.85 | 4.36 | 2.91 | 2.45 | 2.17 | 2.05 |
| UniPC | EDM | 23.52 | 11.10 | 3.86 | 2.85 | 2.38 | 2.08 | 2.01 |
| DPM-Solver-v3 | | 12.21 | 8.56 | 3.50 | 2.51 | 2.24 | 2.10 | 2.02 |
| RE-based | | **11.82** | **8.30** | **3.46** | **2.48** | **2.21** | **2.07** | **2.01** |

CONCLUSIONS

In this paper, we introduce a novel framework that leverages variance-driven conditional entropy reduction to improve the sampling performance of pre-trained diffusion model without the need for retraining. Our theoretical analysis establishes that minimizing conditional entropy in the reverse process of diffusion models leads to more accurate and efficient denoising, which provides a principled foundation for optimizing this process. Building on these insights, we propose a Reduced Entropy (RE) approach for sampling of diffusion models, which improves the denoising process through efficient conditional variance minimization. Our method achieves state-of-the-art performance across multiple benchmark training-free methods, demonstrating promising improvements in both sampling speed and generation quality. While our approach yields promising results in image generation tasks, the full potential of conditional entropy reduction-based sampling methods remains to be explored.

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

APPENDIX

## A    RELATED WORK

### A.1    DIFFUSION MODELS

The mathematical foundation of Diffusion models (DMs) originates from stochastic differential equations (SDEs), particularly the Langevin dynamics Langevin et al. (1908). This framework was first adapted to deep generative modeling in the work of Sohl-Dickstein et al. Sohl-Dickstein et al. (2015) through a Markov chain approach based on non-equilibrium thermodynamics. Significant advances in sample quality were achieved by Song and Ermon Song & Ermon (2019; 2020) through score-based generative models, introducing effective methods to estimate and sample from the score function $\nabla_{\boldsymbol{x}} \log p(\boldsymbol{x})$ of data distributions. The introduction of DDPMs by Ho et al. Ho et al. (2020) provided a significant simplification by parameterizing the reverse process and optimizing a variational bound, establishing a stable training objective: $L = \mathbb{E}_{t,\boldsymbol{\epsilon}}\|\boldsymbol{\epsilon} - \boldsymbol{\epsilon}_\theta(\boldsymbol{x}_t, t)\|^2$. Subsequently, Song et al. Song et al. (2021b) unified score-based models and DMs under a continuous-time SDE formulation, providing a rigorous mathematical foundation.

Building upon this theoretical framework, diffusion models have demonstrated exceptional capabilities across various domains. In image synthesis, they have achieved state-of-the-art quality Dhariwal & Nichol (2021) and established new benchmarks in photorealism Karras et al. (2022). Their success has extended to multimodal generation tasks, including text-to-image synthesis Ramesh et al. (2022); Saharia et al. (2022), speech generation Chen et al. (2021), video synthesis Ho et al. (2022), and 3D content generation Poole et al. (2023). Furthermore, DMs have shown remarkable capabilities in controllable generation tasks Zhang et al. (2023), such as image editing, style transfer, and inpainting Meng et al. (2022); Lugmayr et al. (2022).

### A.2    TRAINING-BASED FAST SAMPLING METHODS

Training-based sampling methods improve the efficiency of diffusion models through learning-based optimization of sampling trajectories. Knowledge distillation techniques facilitate efficient sampling, where Progressive Distillation Salimans & Ho (2022) enables student models to learn compressed sampling processes from teacher models. Consistency-based methods establish another fundamental direction, with Consistency Models Song et al. (2023); Song & Dhariwal (2024) achieving parallel generation through score function learning via consistency training, grounded in probability flow ODE frameworks. Rectified Flow Liu et al. (2023) formulates a rectified flow ODE that enhances sampling efficiency with the paired training strategy. Latent Consistency Models Luo et al. (2023) extend consistency distillation to latent space, maintaining generation quality while accelerating sampling through the integration of latent diffusion and consistency training. Additional developments encompass architectural improvements and training optimizations Nichol & Dhariwal (2021); Dhariwal & Nichol (2021); Zhang & Chen (2021).

Despite the acceleration benefits, these approaches typically necessitate specialized training procedures and careful balancing of quality-speed trade-offs. The training data for these methods are typically obtained through iterative sampling from pre-trained DMs using deterministic samplers like DDIM Song et al. (2021a).

### A.3    TRAINING-FREE SAMPLING METHODS

In contrast, training-free methods focus on improving sampling efficiency without requiring any additional training, making them more adaptable and flexible in nature. Early sampling methods in DMs relied on ancestral sampling Ho et al. (2020). Score-based models Song et al. (2021b) used predictor-corrector methods to refine samples and introduced PF ODEs as a faster sampling alternative. DDIM Song et al. (2021a) advanced sampling methods by introducing a non-Markovian deterministic process that enables deterministic sampling through a variance-minimizing path, significantly reducing the number of required sampling steps. PNDM Liu et al. (2022) demonstrated the adaptability of ODE solvers to diffusion sampling by effectively utilizing linear multistep methods. EDM Karras et al. (2022) explored the design space of DMs with a $\sigma$-parameterization linked to the SNR, analyzing noise dynamics to optimize time steps and achieved high-quality samples using the Heun solver. DPM-Solver Lu et al. (2022a) introduced a generalized acceleration framework leveraging favorable properties of PF ODEs in the semi-log-SNR space, with high-order solvers for accelerated sampling. DEIS Zhang & Chen (2023) investigated the effectiveness of exponential integrators in addressing the stiffness of diffusion ODEs. DPM-Solver++ Lu et al. (2022b) extended DPM-Solver to guided sampling by using data-based parameterization, further improved efficiency.

Based on DPM-Solver, UniPC Zhao et al. (2024) designed high-order predictor-corrector schemes within a unified framework and demonstrated their strengths experimentally. Furthermore, DPM-Solver-v3 Zheng et al. (2023a) proposed an optimized ODE parameterization using empirical model statistics (EMS), which significantly enhanced the sampling efficiency of DMs.

Despite significant progress in the numerical discretization techniques of training-free methods, the underlying mechanisms driving their acceleration remain inadequately understood. This motivates our study, which seeks to unify efficient numerical iterations, such as DPM-Solver and EDM, through the lens of conditional entropy reduction.

## B  ANALYSIS AND PROOFS OF VARIANCE-DRIVEN CONDITIONAL ENTROPY REDUCTION

### B.1  THE PROOF OF PROPOSITION 3.1

*Proof.* Denote the Gaussian transition distributions governed by the iterative equations (3.5) and (3.7) as $p_1\left(\boldsymbol{f}(\tilde{\boldsymbol{x}}_{t_{i-1}})|\boldsymbol{f}(\tilde{\boldsymbol{x}}_{t_i})\right)$ and $p_2\left(\boldsymbol{f}(\tilde{\boldsymbol{x}}_{t_{i-1}})|\boldsymbol{f}(\tilde{\boldsymbol{x}}_{t_i})\right)$, respectively. Without loss of generality, we use the common part $\boldsymbol{f}(\tilde{\boldsymbol{x}}_{t_i})$ of the two iterative equations as the mean of both distributions. The remaining components represent the perturbation terms associated with each transition distribution, respectively. Since the noise prediction model is specifically trained to predict the noise, we can interpret $\boldsymbol{\epsilon}_\theta(\tilde{\boldsymbol{x}}_t, t)$ as representing the noise perturbation term. Since the estimated noise by the model at different time steps can be considered mutually independent, the conditional variances of the remaining terms for the two different iterations are, respectively, expressed as follows:

$$\mathrm{Var}_{p_1} = h_{t_i}^2 \cdot \mathrm{Var}(\boldsymbol{\epsilon}_\theta(\tilde{\boldsymbol{x}}_{t_i}, t_i)), \ \mathrm{Var}_{p_2} = h_{t_i}^2\left(1 - \frac{h_{t_i}}{2\hat{h}_{t_i}}\right)^2 \cdot \mathrm{Var}(\boldsymbol{\epsilon}_\theta(\tilde{\boldsymbol{x}}_{t_i}, t_i)) + \frac{h_{t_i}^4}{4\hat{h}_{t_i}^2} \cdot \mathrm{Var}(\boldsymbol{\epsilon}_\theta(\tilde{\boldsymbol{x}}_{s_i}, s_i)). \quad (\text{B.1})$$

Denote $\Delta H(p) = H_{p_2}(\tilde{\boldsymbol{x}}_{t_{i-1}}|\tilde{\boldsymbol{x}}_{t_i}) - H_{p_1}(\tilde{\boldsymbol{x}}_{t_{i-1}}|\tilde{\boldsymbol{x}}_{t_i})$. Then, by equations (B.1) and (3.10), we have:

$$\Delta H(p) = \frac{d}{2}\log\left|1 - \frac{h_{t_i}}{\hat{h}_{t_i}} + \frac{h_{t_i}^2}{4\hat{h}_{t_i}^2} + \frac{h_{t_i}^2}{4\hat{h}_{t_i}^2} \cdot \frac{\mathrm{Var}(\boldsymbol{\epsilon}_\theta(\tilde{\boldsymbol{x}}_{s_i}, s_i))}{\mathrm{Var}(\boldsymbol{\epsilon}_\theta(\tilde{\boldsymbol{x}}_{t_i}, t_i))}\right|. \quad (\text{B.2})$$

Therefore, $\Delta H(p) \leq 0$ if and only if $\frac{h_{t_i}^2}{4\hat{h}_{t_i}^2} + \frac{h_{t_i}^2}{4\hat{h}_{t_i}^2} \cdot \frac{\mathrm{Var}(\boldsymbol{\epsilon}_\theta(\tilde{\boldsymbol{x}}_{s_i}, s_i))}{\mathrm{Var}(\boldsymbol{\epsilon}_\theta(\tilde{\boldsymbol{x}}_{t_i}, t_i))} \leq \frac{h_{t_i}}{\hat{h}_{t_i}}$. By solving this inequality and note that $\hat{h}_{t_i} \leq h_{t_i}$, the proof is complete. $\square$

### B.2  THE PERSPECTIVE OF CONDITIONAL ENTROPY REDUCTION FOR SOME ACCELERATED ITERATIONS

As an application of conditional entropy analysis, we deepen our understanding of the iterations in accelerated denoising diffusion solvers, such as DPM-Solver Lu et al. (2022a) and EDM Karras et al. (2022), by elucidating the associated changes in conditional entropy. We then demonstrate that the iterations of both well-known solvers are denoising iterations grounded in conditional entropy reduction and represent two special cases of RE-based iterations.

Firstly, let us revisit the accelerated iteration introduced by EDM Karras et al. (2022). Formally, the iteration formula of EDM can be written as follows:

$$\boldsymbol{f}(\tilde{\boldsymbol{x}}_{t_{i-1}}) = \boldsymbol{f}(\tilde{\boldsymbol{x}}_{t_i}) + h_{t_i}\frac{\boldsymbol{d}_\theta\left(\tilde{\boldsymbol{x}}_{t_i}, t_i\right) + \boldsymbol{d}_\theta\left(\tilde{\boldsymbol{x}}_{t_{i-1}}, t_{i-1}\right)}{2}, \quad (\text{B.3})$$

which can be equivalently rewritten as the following gradient estimation-based iteration:

$$\boldsymbol{f}(\tilde{\boldsymbol{x}}_{t_{i-1}}) = \boldsymbol{f}(\tilde{\boldsymbol{x}}_{t_i}) + h_{t_i}\boldsymbol{d}_\theta\left(\tilde{\boldsymbol{x}}_{t_i}, t_i\right) + \frac{h_{t_i}^2}{2}F_\theta(t_{i-1}, t_i). \quad (\text{B.4})$$

As $\hat{h}_{t_i} = h_{t_i}$ in the iteration of EDM described by Eq. B.4, based on Proposition 3.2, we obtain the following conclusion:

**Remark 3** *The EDM iteration in Eq. (B.3) can reduce conditional entropy more effectively than the DDIM iteration in Eq. (3.5). Thus, the iteration of EDM can be interpreted as an iterative scheme for reducing conditional entropy.*

Next, we revisit the accelerated iteration framework established by DPM-Solver Lu et al. (2022a) with exponential integrator. Specifically, the sampling algorithm of DPM-Solver decouples the semi-linear structure of the diffusion ODE, with its iterations formulated by solving the integral driven by half of the log-SNR. The exponentially weighted score integral in DPM-Solver can be written as follows:

$$\boldsymbol{f}(\boldsymbol{x}_t) - \boldsymbol{f}(\boldsymbol{x}_s) = -\int_{\lambda(s)}^{\lambda(t)} e^{-\tau}\boldsymbol{d}_\theta\left(\boldsymbol{x}_{\psi(\tau)}, \psi(\tau)\right)\mathrm{d}\tau. \quad (\text{B.5})$$

where $\lambda(t) := \log \frac{\alpha_t}{\sigma_t}$. It follows that Eq. (B.5) and Eq. (3.2) can be mutually transformed through the function relation $\lambda(t) = -\log(\kappa(t))$. Denote $h_{\lambda_i} := \lambda(t_{i-1}) - \lambda(t_i)$ and $\hat{h}_{\lambda_i} := \lambda(s_i) - \lambda(t_i)$. Formally, the second-order iteration of DPM-Solver can be written as follows:

$$\boldsymbol{f}(\tilde{\boldsymbol{x}}_{t_{i-1}}) = \boldsymbol{f}(\tilde{\boldsymbol{x}}_{t_i}) - \frac{\sigma_{t_{i-1}}}{\alpha_{t_{i-1}}}\left(e^{h_{\lambda_i}} - 1\right)\boldsymbol{d}_\theta\left(\tilde{\boldsymbol{x}}_{t_i}, t_i\right) - \frac{\sigma_{t_{i-1}}}{\alpha_{t_{i-1}}}\left(e^{h_{\lambda_i}} - 1\right)\frac{\boldsymbol{d}_\theta\left(\tilde{\boldsymbol{x}}_{s_i}, s_i\right) - \boldsymbol{d}_\theta\left(\tilde{\boldsymbol{x}}_{t_i}, t_i\right)}{2r_1}, \quad \text{(B.6)}$$

where $s_i = \psi\left(\lambda(t_i) + r_1 h_{\lambda_i}\right)$. Note that $r_1 = \frac{\hat{h}_{\lambda_i}}{h_{\lambda_i}}$, $\kappa(t_{i-1}) = \frac{\sigma_{t_{i-1}}}{\alpha_{t_{i-1}}}$ and $h_{t_i} = \kappa(t_{i-1}) - \kappa(t_i)$. As $e^{h_{\lambda_i}} = \frac{\kappa(t_i)}{\kappa(t_{i-1})}$, then $\frac{\sigma_{t_{i-1}}}{\alpha_{t_{i-1}}}\left(e^{h_{\lambda_i}} - 1\right) = -h_{t_i}$. Thus, this second-order iteration can be equivalently rewritten as:

$$\boldsymbol{f}(\tilde{\boldsymbol{x}}_{t_{i-1}}) = \boldsymbol{f}(\tilde{\boldsymbol{x}}_{t_i}) + h_{t_i}\boldsymbol{d}_\theta\left(\tilde{\boldsymbol{x}}_{t_i}, t_i\right) + \frac{h_{t_i}h_{\lambda_i}}{2}\frac{\boldsymbol{d}_\theta\left(\tilde{\boldsymbol{x}}_{s_i}, s_i\right) - \boldsymbol{d}_\theta\left(\tilde{\boldsymbol{x}}_{t_i}, t_i\right)}{\hat{h}_{\lambda_i}}. \quad \text{(B.7)}$$

Note that the $s_i$ here in DPM-Solver differs from the one in Eq. (3.7), due to the variations arising from the function space. Based on conditional analysis, similarly, we have the following conclusion.

**Remark 4** *Based on Proposition 3.2, when $\frac{h_{\lambda_i}}{\hat{h}_{\lambda_i}} \in \left[1, \frac{4\,\text{SNR}(s_i)}{\text{SNR}(t_i) + \text{SNR}(s_i)}\right]$, the DPM-Solver's iteration in Eq. (B.6) can reduce conditional entropy more effectively than the DDIM iteration in Eq. (3.5). Note that $\frac{h_{\lambda_i}}{\hat{h}_{\lambda_i}} = 2$ in the practical implementation of DPM-Solver. Thus, as $\text{SNR}(s_i) > \text{SNR}(t_i)$, the DPM-Solver's iteration can be interpreted as an iterative scheme for reducing conditional entropy.*

Finally, we summarize the relationship between these two iterations and RE-based iterations. In fact, the iteration described in Eq. (B.3) is an RE-based iteration within the EDM iteration framework. Clearly, the RE-based iteration within the DPM-Solver iteration framework can be formulated as:

$$\boldsymbol{f}(\tilde{\boldsymbol{x}}_{t_{i-1}}) = \boldsymbol{f}(\tilde{\boldsymbol{x}}_{t_i}) + h_{t_i}\left(\gamma\boldsymbol{d}_\theta\left(\tilde{\boldsymbol{x}}_{s_i}, s_i\right) + (1-\gamma)\boldsymbol{d}_\theta\left(\tilde{\boldsymbol{x}}_{t_i}, t_i\right)\right) + \frac{h_{t_i}h_{\lambda_i}}{2}\frac{\boldsymbol{d}_\theta\left(\tilde{\boldsymbol{x}}_{s_i}, s_i\right) - \boldsymbol{d}_\theta\left(\tilde{\boldsymbol{x}}_{t_i}, t_i\right)}{\hat{h}_{\lambda_i}}. \quad \text{(B.8)}$$

Therefore, the iterations in both EDM and DPM-Solver can be interpreted as specific instances of RE-based denoising iterations from the perspective of the conditional entropy.

### B.3 Difference Analysis of Gradient Estimation-based Iterations in Multi-step Framework

On one hand, let us revisit the multi-step accelerated framework established by DPM-Solver++ Lu et al. (2022b). Formally, the second-order iteration of DPM-Solver++ can be written as follows

$$\boldsymbol{f}(\tilde{\boldsymbol{x}}_{t_{i-1}}) = \boldsymbol{f}(\tilde{\boldsymbol{x}}_{t_i}) - \frac{\alpha_{t_{i-1}}}{\sigma_{t_{i-1}}}\left(e^{-h_{\lambda_i}} - 1\right)\boldsymbol{d}_\theta\left(\tilde{\boldsymbol{x}}_{t_i}, t_i\right) - \frac{\alpha_{t_{i-1}}}{\sigma_{t_{i-1}}}\left(e^{-h_{\lambda_i}} - 1\right)\frac{\boldsymbol{d}_\theta\left(\tilde{\boldsymbol{x}}_{t_i}, t_i\right) - \boldsymbol{d}_\theta\left(\tilde{\boldsymbol{x}}_{t_{i+1}}, t_{i+1}\right)}{2r_i}, \quad \text{(B.9)}$$

where $\boldsymbol{d}_\theta\left(\tilde{\boldsymbol{x}}_{t_i}, t_i\right)$ denotes the data-prediction prediction model and $r_i = \frac{h_{\lambda_{i+1}}}{h_{\lambda_i}}$. Since $\frac{\alpha_{t_{i-1}}}{\sigma_{t_{i-1}}}\left(e^{-h_{\lambda_i}} - 1\right) = \frac{\alpha_{t_i}}{\sigma_{t_i}} - \frac{\alpha_{t_{i-1}}}{\sigma_{t_{i-1}}} = -h_{t_i}$ in data-prediction prediction models, Eq. (B.9) can be rewritten as

$$\boldsymbol{f}(\tilde{\boldsymbol{x}}_{t_{i-1}}) = \boldsymbol{f}(\tilde{\boldsymbol{x}}_{t_i}) + h_{t_i}\boldsymbol{d}_\theta\left(\tilde{\boldsymbol{x}}_{t_i}, t_i\right) + \frac{h_{t_i}h_{\lambda_i}}{2}\frac{\boldsymbol{d}_\theta\left(\tilde{\boldsymbol{x}}_{t_i}, t_i\right) - \boldsymbol{d}_\theta\left(\tilde{\boldsymbol{x}}_{t_{i+1}}, t_{i+1}\right)}{h_{\lambda_{i+1}}}. \quad \text{(B.10)}$$

On the other hand, we can rewrite the iteration presented in Eq. (4.4) as follows:

$$\boldsymbol{f}(\tilde{\boldsymbol{x}}_{t_{i-1}}) = \boldsymbol{f}(\tilde{\boldsymbol{x}}_{t_i}) + h_{t_i}\boldsymbol{d}_\theta\left(\tilde{\boldsymbol{x}}_{t_i}, t_i\right) + \frac{h_{t_i}^2}{2}\frac{\boldsymbol{d}_\theta\left(\tilde{\boldsymbol{x}}_{t_i}, t_i\right) - \boldsymbol{d}_\theta\left(\tilde{\boldsymbol{x}}_{t_{i+1}}, t_{i+1}\right)}{h_{t_{i+1}}}. \quad \text{(B.11)}$$

It has been observed that the differences in the multi-step iterations presented in Eq. (B.10) and Eq. (B.11) are still caused by the variations in $r_i$. Therefore, in gradient estimation-based iterations, the core characteristic of the DPM-Solver++ iteration is the determination of $r_i$ in the half-logarithmic SNR space. For convenience, we will hereafter refer to half-logarithmic SNR simply as 'logSNR'. Without loss of generality, the core differences between various gradient estimation-based iterations can be generalized as variations in the determination of $r_i$. Then, a natural question arises: how can $r_i$ be determined better or systematically? Therefore, a principle for determining $r_i$ is of great importance. This inquiry drives our investigation from the perspective of conditional entropy within the context of multi-step iterations.

### B.4 THE PROOF OF PROPOSITION 3.4 FOR CONDITIONAL ENTROPY REDUCTION

*Proof.* Without loss of generality, we only need to prove that the conditional entropy of the first-order iteration using data-prediction parameterization is lower than that of the first-order iteration using noise-prediction parameterization. Let us revisit both first-order denoising iterations. Clearly, based on Eq. (3.5) and Remark 1, the first-order iteration of data-prediction parameterization as follows:

$$\tilde{x}_{t_{i-1}} = \underbrace{\frac{\sigma_{t_{i-1}}}{\sigma_{t_i}} \tilde{x}_{t_i}}_{L_{\text{data}}: \text{ linear}} + \underbrace{\sigma_{t_{i-1}} \left( \frac{\alpha_{t_{i-1}}}{\sigma_{t_{i-1}}} - \frac{\alpha_{t_i}}{\sigma_{t_i}} \right) x_\theta (\tilde{x}_{t_i}, t_i)}_{N_{\text{data}}: \text{ non-linear}}, \tag{B.12}$$

where $x_\theta (\tilde{x}_{t_i}, t_i) = \frac{\tilde{x}_{t_i} - \sigma_{t_i} \epsilon_\theta (\tilde{x}_{t_i}, t_i)}{\alpha_{t_i}}$. The first-order iteration of noise-prediction parameterization as follows:

$$\tilde{x}_{t_{i-1}} = \underbrace{\frac{\alpha_{t_{i-1}}}{\alpha_{t_i}} \tilde{x}_{t_i}}_{L_{\text{noise}}: \text{ linear}} + \underbrace{\alpha_{t_{i-1}} \left( \frac{\sigma_{t_{i-1}}}{\alpha_{t_{i-1}}} - \frac{\sigma_{t_i}}{\alpha_{t_i}} \right) \epsilon_\theta (\tilde{x}_{t_i}, t_i)}_{N_{\text{noise}}: \text{ non-linear}}. \tag{B.13}$$

Denote the Gaussian transition kernels governed by the iterative equations (B.12) and (B.13) as $p_1 (\tilde{x}_{t_{i-1}} | x_0)$ and $p_2 (\tilde{x}_{t_{i-1}} | x_0)$, respectively. In both iterations of equations (B.12) and (B.13), the randomness of the linear term is solely related to the noise introduced in the iterations preceding time step $t_i$, whereas the randomness of the nonlinear term depends entirely on the noise at the current time step $t_i$. Since the noise introduced at each time step in DMs is independent, under this assumption, the randomness of the linear term is independent of the randomness of the nonlinear term in both iterations. Therefore, we will consider the variances of the linear and nonlinear components separately. Formally, the variances of the linear terms for the two different iterations are, respectively, as follows:

$$\text{Var}(L_{\text{data}} | x_0) = \frac{\sigma_{t_{i-1}}^2}{\sigma_{t_i}^2} \text{Var}(\tilde{x}_{t_i} | x_0), \quad \text{Var}(L_{\text{noise}} | x_0) = \frac{\alpha_{t_{i-1}}^2}{\alpha_{t_i}^2} \text{Var}(\tilde{x}_{t_i} | x_0). \tag{B.14}$$

For simplicity, we denote $\text{Var}(\tilde{x}_{t_i} | x_0)$ as $\text{Var}(\tilde{x}_{t_i})$ where appropriate. Based on monotonicity, $\frac{\sigma_{t_{i-1}}}{\sigma_{t_i}} < \frac{\alpha_{t_{i-1}}}{\alpha_{t_i}}$, as $\alpha_t$ is monotonically decreasing with respect to time $t$ and $\sigma_t$ is monotonically increasing with respect to time $t$. Therefore, $\text{Var}(L_{\text{data}}) < \text{Var}(L_{\text{noise}})$. Subsequently, we consider the variance of the non-linear terms for both iterations. For clarity, we denote $c(t_i, t_{i-1}) := \alpha_{t_i} \sigma_{t_{i-1}} - \alpha_{t_{i-1}} \sigma_{t_i}$. Then,

$$\sigma_{t_{i-1}} \left( \frac{\alpha_{t_{i-1}}}{\sigma_{t_{i-1}}} - \frac{\alpha_{t_i}}{\sigma_{t_i}} \right) = \frac{-1}{\sigma_{t_i}} c(t_i, t_{i-1}), \quad \alpha_{t_{i-1}} \left( \frac{\sigma_{t_{i-1}}}{\alpha_{t_{i-1}}} - \frac{\sigma_{t_i}}{\alpha_{t_i}} \right) = \frac{1}{\alpha_{t_i}} c(t_i, t_{i-1}). \tag{B.15}$$

Thus, the variances of the nonlinear terms for the two different iterations are, respectively, as follows:

$$\text{Var}(N_{\text{noise}}) = \frac{c^2(t_i, t_{i-1})}{\alpha_{t_i}^2} \cdot \text{Var}(\epsilon_\theta (\tilde{x}_{t_i}, t_i)), \quad \text{Var}(N_{\text{data}}) = \frac{(-c(t_i, t_{i-1}))^2}{\sigma_{t_i}^2} \cdot \text{Var}(x_\theta (\tilde{x}_{t_i}, t_i)). \tag{B.16}$$

Note that

$$\text{Var}(x_\theta (\tilde{x}_{t_i}, t_i)) = \text{Var}\left( \frac{\tilde{x}_{t_i} - \sigma_{t_i} \epsilon_\theta (\tilde{x}_{t_i}, t_i)}{\alpha_{t_i}} \right) = \frac{\sigma_{t_i}^2}{\alpha_{t_i}^2} \text{Var}(\epsilon_{t_i} - \epsilon_\theta (\tilde{x}_{t_i}, t_i)). \tag{B.17}$$

as $\tilde{x}_{t_i} = \alpha_{t_i} x_0 + \sigma_{t_i} \epsilon_{t_i}$. Then,

$$\text{Var}(N_{\text{data}}) = \frac{(-c(t_i, t_{i-1}))^2}{\sigma_{t_i}^2} \cdot \frac{\sigma_{t_i}^2}{\alpha_{t_i}^2} \text{Var}(\epsilon_{t_i} - \epsilon_\theta (\tilde{x}_{t_i}, t_i)) = \frac{c^2(t_i, t_{i-1})}{\alpha_{t_i}^2} \text{Var}(\epsilon_\theta (\tilde{x}_{t_i}, t_i) - \epsilon_{t_i}). \tag{B.18}$$

Clearly, since $\epsilon_\theta (\tilde{x}_{t_i}, t_i)$ is designed to predict the injected noise into the clean data at time step $t_i$, and based on Eq. (2.4), the variance $\text{Var}(\epsilon_\theta (\tilde{x}_{t_i}, t_i) - \epsilon_{t_i})$ can theoretically approach arbitrarily small values as the accuracy of the model's estimation improves. Therefore, as $\text{Var}(\epsilon_\theta (\tilde{x}_{t_i}, t_i) - \epsilon_{t_i}) < \text{Var}(\epsilon_\theta (\tilde{x}_{t_i}, t_i))$, we have $\text{Var}(N_{\text{data}}) < \text{Var}(N_{\text{niose}})$. Since the randomness of the linear term is independent of that of the nonlinear term in both iterations, and given that $\text{Var}(L_{\text{data}}) < \text{Var}(L_{\text{noise}})$ and $\text{Var}(N_{\text{data}}) < \text{Var}(N_{\text{niose}})$, we have

$$0 \le \text{Var}(p_1(\tilde{x}_{t_{i-1}} | x_0)) = \text{Var}(L_{\text{data}}) + \text{Var}(N_{\text{data}}) < \text{Var}(L_{\text{noise}}) + \text{Var}(N_{\text{noise}}) = \text{Var}(p_2(\tilde{x}_{t_{i-1}} | x_0)). \tag{B.19}$$

Consequently, based on Eq. (3.10), which provides the conditional entropy formula for a Gaussian distribution, we have $H_{p_1}(\tilde{x}_{t_{i-1}} | x_0) < H_{p_2}(\tilde{x}_{t_{i-1}} | x_0)$. The proof is complete. □

## C  CONVERGENCE ANALYSIS OF RE-BASED ITERATIVE METHODS

### C.1  ASSUMPTION

**Assumption 1**: The total derivative $d_\theta^{(k)}\left(x_{\psi(\tau)}, \psi(\tau)\right) := \frac{\mathrm{d}^k d_\theta(x_{\psi(\tau)}, \psi(\tau))}{\mathrm{d}\,\tau^k}$ exists and is continuous if necessary, where $k$ is determined by the specific context.

**Assumption 2**: The function $d_\theta\left(x_{\psi(\tau)}, \psi(\tau)\right)$ is Lipschitz w.r.t. to its first parameter $x_{\psi(\tau)}$.

### C.2  THE PROOF OF THEOREM 4.1 FOR CONVERGENCE ORDER

Let us review the RE-based iteration as follows:

$$f(\tilde{x}_{t_{i-1}}) = f(\tilde{x}_{t_i}) + h_{t_i}\left(\gamma d_\theta\left(\tilde{x}_{s_i}, s_i\right) + (1-\gamma)d_\theta\left(\tilde{x}_{t_i}, t_i\right)\right) + \frac{h_{t_i}^2}{2}F_\theta(s_i, t_i),$$

where $F_\theta(s_i, t_i) := \frac{d_\theta(\tilde{x}_{s_i}, s_i) - d_\theta(\tilde{x}_{t_i}, t_i)}{\hat{h}_{t_i}}$, $\hat{h}_{t_i} := \kappa(s_i) - \kappa(t_i)$.

*Proof.* Denotes $\hat{x}_t = f(\tilde{x}_t)$. The RE-based iteration can be decomposed into:

$$\hat{x}_\mu = \hat{x}_{t_i} + h_{t_i} d_\theta\left(\tilde{x}_{t_i}, t_i\right) + \frac{h_{t_i}^2}{2}F_\theta(s_i, t_i),$$

and

$$\hat{x}_{t_{i-1}} = \hat{x}_\mu + \gamma h_{t_i}\left(d_\theta\left(\tilde{x}_{s_i}, s_i\right) - d_\theta\left(\tilde{x}_{t_i}, t_i\right)\right).$$

Clearly, $\hat{x}_\mu = \hat{x}_{t_i} + O(h_{t_i}^3)$ based on the Taylor expansion in Eq. (3.4). Since the model $d_\theta\left(\tilde{x}_t, t\right)$ satisfies the Lipschitz assumption with respect to $\tilde{x}_t$, then

$$\begin{aligned}
\|\hat{x}_{t_{i-1}} - \hat{x}_\mu\| &= \|\gamma h_{t_i}\left(d_\theta\left(\tilde{x}_{s_i}, s_i\right) - d_\theta\left(\tilde{x}_{t_i}, t_i\right)\right)\| \\
&= L_1 \hat{h}_{t_i}\|d_\theta\left(\tilde{x}_{s_i}, s_i\right) - d_\theta\left(\tilde{x}_{t_i}, t_i\right)\| \\
&\leq L_2 \hat{h}_{t_i}\|\tilde{x}_{s_i} - \tilde{x}_{t_i}\| = O(|\hat{h}_{t_i}|^3).
\end{aligned} \tag{C.1}$$

Subsequently, by the triangle inequality, we have

$$\|\hat{x}_{t_{i-1}} - \hat{x}_{t_i}\| = \|\hat{x}_{t_{i-1}} - \hat{x}_\mu + \hat{x}_\mu - \hat{x}_{t_i}\| \leq \|\hat{x}_{t_{i-1}} - \hat{x}_\mu\| + \|\hat{x}_\mu - \hat{x}_{t_i}\| = O(|h_{t_i}|^3), \tag{C.2}$$

where the last equality holds because $|h_{t_i}| \geq |\hat{h}_{t_i}|$.

Therefore, we prove that the local error of the RD-based iteration is of the same order as the corresponding Taylor expansion. Consequently, the RE-based iteration in Eq. 4.1 is a second-order convergence algorithm. The proof is complete.  □

### C.3  THE PROOFS OF LEMMA 4.1 AND LEMMA 4.2

*Proof.* Without loss of generality,

$$\begin{aligned}
&\frac{\partial \|A - \sigma h\left(\lambda F_1 + (1-\lambda)F_2\right)\|_F^2}{\partial \lambda} \\
&= \frac{\partial\left(\mathrm{vec}^\top\left(A - \sigma h\left(\lambda F_1 + (1-\lambda)F_2\right)\right)\mathrm{vec}\left(A - \sigma h\left(\lambda F_1 + (1-\lambda)F_2\right)\right)\right)}{\partial \lambda} \\
&= 2\,\mathrm{vec}^\top\left(\frac{\partial\left(A - \sigma h\left(\lambda F_1 + (1-\lambda)F_2\right)\right)}{\partial \lambda}\right)\mathrm{vec}\left(A - \sigma h\left(\lambda F_1 + (1-\lambda)F_2\right)\right) \\
&= 2\,\mathrm{vec}^\top\left(-\sigma h\left(F_1 - F_2\right)\right)\mathrm{vec}\left(A - \sigma h\left(\lambda F_1 + (1-\lambda)F_2\right)\right)
\end{aligned}$$

Let $\frac{\partial \|A - \sigma h(\lambda F_1 + (1-\lambda)F_2)\|_F^2}{\partial \lambda} = 0$, we have

$$\mathrm{vec}^\top\left(F_1 - F_2\right)\mathrm{vec}\left(\sigma h\lambda\left(F_1 - F_2\right) - \left(A - \sigma h F_2\right)\right) = 0. \tag{C.3}$$

Therefore,

$$\lambda = \frac{\mathrm{vec}^\top\left(F_1 - F_2\right)\mathrm{vec}\left(A - \sigma h F_2\right)}{\sigma h\,\mathrm{vec}^\top\left(F_1 - F_2\right)\mathrm{Vec}\left(F_1 - F_2\right)}. \tag{C.4}$$

The proof is complete.  □

## C.4 THE CONVERGENCE ANALYSIS OF RE-BASED ITERATION IN ALGORITHM 1

Let us review the RE-based iteration in Algorithm 1 as follows:

$$
\begin{aligned}
\boldsymbol{f}(\tilde{\boldsymbol{x}}_{t_{i-1}}) &= \boldsymbol{f}(\tilde{\boldsymbol{x}}_{t_i}) + h_{t_i} \boldsymbol{d}_\theta\left(\tilde{\boldsymbol{x}}_{t_i}, t_i\right) + \frac{h_{t_i}^2}{2} \zeta_i B_\theta(t_i) \\
&= \boldsymbol{f}(\tilde{\boldsymbol{x}}_{t_i}) + h_{t_i} \boldsymbol{d}_\theta\left(\tilde{\boldsymbol{x}}_{t_i}, t_i\right) + \frac{h_{t_i}^2}{2} \zeta_i \left(\frac{\eta_i}{2} B_\theta(s_i, t_i) + \left(1 - \frac{\eta_i}{2}\right) B_\theta(t_i, l_i)\right) \\
&= \boldsymbol{f}(\tilde{\boldsymbol{x}}_{t_i}) + h_{t_i} \left(\frac{\eta_i}{2} \boldsymbol{d}_\theta\left(\tilde{\boldsymbol{x}}_{t_i}, t_i\right) + \left(1 - \frac{\eta_i}{2}\right) \boldsymbol{d}_\theta\left(\tilde{\boldsymbol{x}}_{t_i}, t_i\right)\right) + \frac{h_{t_i}^2}{2} \zeta_i \left(\frac{\eta_i}{2} B_\theta(s_i, t_i) + \left(1 - \frac{\eta_i}{2}\right) B_\theta(t_i, l_i)\right),
\end{aligned}
$$

where $B_\theta(t_i, t_{i+1}) = \frac{\boldsymbol{d}_\theta(\tilde{\boldsymbol{x}}_{t_i}, t_i) - \boldsymbol{d}_\theta(\tilde{\boldsymbol{x}}_{t_{i+1}}, t_{i+1})}{h_{t_{i+1}}}$.

We now analyze the convergence properties of this RE-based iteration scheme and establish its convergence order. We have the following result.

**Theorem C.1** *The RE-based iteration in Algorithm 1 achieves second-order global convergence with local error $O(h_{t_i}^3)$.*

*Proof.* Denote $\hat{\boldsymbol{x}}_t = \boldsymbol{f}(\tilde{\boldsymbol{x}}_t)$. The RE-based iteration in Algorithm 1 can be decomposed as:

$$
\begin{aligned}
\hat{\boldsymbol{x}}_{t_{i-1}} &= \hat{\boldsymbol{x}}_{t_i} + \frac{\eta_i}{2} \hat{\boldsymbol{x}}_{\mu_1} + \left(1 - \frac{\eta_i}{2}\right) \hat{\boldsymbol{x}}_{\mu_2} \\
&= \frac{\eta_i}{2} \left(\hat{\boldsymbol{x}}_{t_i} + \hat{\boldsymbol{x}}_{\mu_1}\right) + \left(1 - \frac{\eta_i}{2}\right) \left(\hat{\boldsymbol{x}}_{t_i} + \hat{\boldsymbol{x}}_{\mu_2}\right),
\end{aligned}
$$

where

$$
\hat{\boldsymbol{x}}_{\mu_1} = h_{t_i} \boldsymbol{d}_\theta(\tilde{\boldsymbol{x}}_{t_i}, t_i) + \frac{h_{t_i}^2}{2} \zeta_i B_\theta(s_i, t_i), \quad \hat{\boldsymbol{x}}_{\mu_2} = h_{t_i} \boldsymbol{d}_\theta(\tilde{\boldsymbol{x}}_{t_i}, t_i) + \frac{h_{t_i}^2}{2} \zeta_i B_\theta(t_i, l_i).
$$

Let us now consider the case of $\hat{\boldsymbol{x}}_{t_i} + \hat{\boldsymbol{x}}_{\mu_1}$. Denote

$$
\hat{\boldsymbol{x}}_{1,t_{i-1}} = \hat{\boldsymbol{x}}_{t_i} + \hat{\boldsymbol{x}}_{\mu_1}, \quad \hat{\boldsymbol{x}}_{\mu_3} = \hat{\boldsymbol{x}}_{t_i} + h_{t_i} \boldsymbol{d}_\theta(\tilde{\boldsymbol{x}}_{t_i}, t_i) + \frac{h_{t_i}^2}{2} B_\theta(s_i, t_i).
$$

Then $\hat{\boldsymbol{x}}_{1,t_{i-1}} = \hat{\boldsymbol{x}}_{\mu_3} + (\zeta_i - 1)\frac{h_{t_i}^2}{2} B_\theta(s_i, t_i)$. Note that $\hat{\boldsymbol{x}}_{\mu_3} = \hat{\boldsymbol{x}}_{t_i} + O(h_{t_i}^3)$ and $B_\theta(s_i, t_i) = O(h_{t_i})$ based on the Taylor expansion in Eq. (3.4). Therefore, we have

$$
\begin{aligned}
\|\hat{\boldsymbol{x}}_{1,t_{i-1}} - \hat{\boldsymbol{x}}_{t_i}\| &= \left\| \hat{\boldsymbol{x}}_{\mu_3} - \hat{\boldsymbol{x}}_{t_i} + \frac{\zeta_i - 1}{2} h_{t_i}^2 B_\theta(s_i, t_i) \right\| \\
&\le \|\hat{\boldsymbol{x}}_{\mu_3} - \hat{\boldsymbol{x}}_{t_i}\| + \left\| \frac{\zeta_i - 1}{2} h_{t_i}^2 B_\theta(s_i, t_i) \right\| \\
&= O(h_{t_i}^3) + L_1 O(h_{t_i}^3) = O(h_{t_i}^3),
\end{aligned} \tag{C.5}
$$

where $L_1$ is a constant because $\zeta_i$ can be bounded by 1. Denote $\hat{\boldsymbol{x}}_{2,t_{i-1}} = \hat{\boldsymbol{x}}_{t_i} + \hat{\boldsymbol{x}}_{\mu_2}$. Symmetrically, we obtain

$$
\|\hat{\boldsymbol{x}}_{2,t_{i-1}} - \hat{\boldsymbol{x}}_{t_i}\| = O(h_{t_i}^3). \tag{C.6}
$$

Now, combining the results, we obtain

$$
\begin{aligned}
\|\hat{\boldsymbol{x}}_{t_{i-1}} - \hat{\boldsymbol{x}}_{t_i}\| &= \left\| \frac{\eta_i}{2} \left(\hat{\boldsymbol{x}}_{1,t_{i-1}} - \hat{\boldsymbol{x}}_{t_i}\right) + \left(1 - \frac{\eta_i}{2}\right) \left(\hat{\boldsymbol{x}}_{2,t_{i-1}} - \hat{\boldsymbol{x}}_{t_i}\right) \right\| \\
&\le \frac{\eta_i}{2} \|\hat{\boldsymbol{x}}_{1,t_{i-1}} - \hat{\boldsymbol{x}}_{t_i}\| + \left(1 - \frac{\eta_i}{2}\right) \|\hat{\boldsymbol{x}}_{2,t_{i-1}} - \hat{\boldsymbol{x}}_{t_i}\| \\
&= \frac{\eta_i}{2} O(h_{t_i}^3) + \left(1 - \frac{\eta_i}{2}\right) O(h_{t_i}^3) = O(h_{t_i}^3).
\end{aligned} \tag{C.7}
$$

Thus, we have shown that the local error of the RE-based iteration in Algorithm 1 is $O(h_{t_i}^3)$. Consequently, the RE-based iteration in Algorithm 1 achieves second-order global convergence. The proof is complete. □

## D    EXPERIMENT DETAILS

In our experiments, we utilize several standard pre-trained models. Specifically, we employ the discrete denoising diffusion probabilistic model Ho et al. (2020), the continuous score-based model Song et al. (2021b), and the uncond EDM model Karras et al. (2022), all trained on CIFAR-10 Krizhevsky (2009). For larger-scale evaluations on high-dimensional data, we adopt the pre-trained models trained on ImageNet dataset Deng et al. (2009) from the baseline method Dhariwal & Nichol (2021). Additionally, we use the pre-trained Latent Diffusion Model and Stable Diffusion model Rombach et al. (2021), where the latter is trained on the LAION-5B dataset Schuhmann et al. (2022) using CLIP Radford et al. (2021) text embeddings as conditioning signals.

### D.1    EXPERIMENTAL COMPUTATIONAL RESOURCES AND DATA

All experiments were conducted on NVIDIA GPUs. For high-dimensional datasets like ImageNet, we utilized the NVIDIA GeForce RTX 3090 GPU with 24GB VRAM, experiments were performed on NVIDIA TITAN X (Pascal) with 12GB VRAM. To ensure fair comparison with prior work, we maintained consistent pre-trained models and experimental settings across both scenarios. We list some of the datasets and codes used in Table 4.

Table 4: Some of the datasets and codes used.

| Name | URL |
| --- | --- |
| CIFAR10 | https://www.cs.toronto.edu/ kriz/cifar.html |
| LSUN-Bedroom | https://www.yf.io/p/lsun |
| ImageNet-256 | https://www.image-net.org |
| ScoreSDE | https://github.com/yang-song/score_sde_pytorch |
| EDM | https://github.com/NVlabs/edm |
| Guided-Diffusion | https://github.com/openai/guided-diffusion |
| Latent-Diffusion | https://github.com/CompVis/latent-diffusion |
| Stable-Diffusion | https://github.com/CompVis/stable-diffusion |
| DPM-Solver | https://github.com/LuChengTHU/dpm-solver |
| DPM-Solver++ | https://github.com/LuChengTHU/dpm-solver |
| UniPC | https://github.com/wl-zhao/UniPC |
| DPM-Solver-v3 | https://github.com/thu-ml/DPM-Solver-v3 |

### D.2    SAMPLING SCHEDULES

Sampling schedules in DMs define how the noise scale evolves during inference and play a crucial role in balancing sample quality and computational efficiency. Several widely used schedules include the Time-uniform schedule Ho et al. (2020); Song et al. (2021b), the LogSNR schedule Lu et al. (2022a), and the EDM schedule Karras et al. (2022). Although optimized schedules have been proposed Xue et al. (2024); Sabour et al. (2024), they typically require significant computational resources for optimization. In our experiments, we follow the default schedule of the baseline methods.

### D.3    PARAMETERIZATION SETTINGS OF THE SAMPLING PROCESS

In the sampling process of DMs, various parameterization settings are used to define the target prediction at each iteration step. Below, we list the adopted parameterizations:
*Noise prediction parameterization* Ho et al. (2020): This parameterization directly predicts the noise injected during the forward diffusion process. The connection to the score function is formalized as:

$$\epsilon_\theta (x_t, t) = -\sigma_t \nabla_x \log q_t (x_t), \tag{D.1}$$

where $\nabla_x \log q_t (x_t)$ denotes the score function Song et al. (2021b).
*Data prediction parameterization* Kingma et al. (2021): This parameterization estimates the clean data $x_0$ from the noisy input $x_t$ at a given time step $t$. The predicted data satisfies:

$$x_\theta(x_t, t) = \frac{x_t - \sigma_t \epsilon_\theta(x_t, t)}{\alpha_t}. \tag{D.2}$$

Although these parameterizations have practical predictive value, they may insufficiently minimize discretization errors. Building upon earlier DPM-Solver versions Lu et al. (2022a;b), DPM-Solver-v3

Zheng et al. (2023a) extends the parameterization strategy by incorporating empirical model statistics (EMS). The continuous-time formulation is as follows:

$$\frac{\mathrm{d}\boldsymbol{x}_\lambda}{\mathrm{d}\lambda} = \left(\frac{\dot{\alpha}_\lambda}{\alpha_\lambda} - \boldsymbol{l}_\lambda\right)\boldsymbol{x}_\lambda - (\sigma_\lambda\boldsymbol{\epsilon}_\theta(\boldsymbol{x}_\lambda, \lambda) - \boldsymbol{l}_\lambda\boldsymbol{x}_\lambda), \tag{D.3}$$

where $\lambda$ represents the continuous-time parameter, and $\boldsymbol{l}_\lambda$ is an optimized prior statistics term.
In this paper, we employ the default parameterization of the baseline method in all of our experiments.

### D.4 Evaluating Sampling Efficiency and Image Quality in Generative Models

The *Fréchet Inception Distance (FID)* Heusel et al. (2017) evaluates the quality and diversity of generated images by comparing the statistical distributions of generated and real images in a feature space. It uses a pre-trained Inception-v3 network to extract features Szegedy et al. (2016), computing the mean $\mu$ and covariance $\Sigma$ for both distributions. Specifically, $\mu_g$ and $\Sigma_g$ represent the mean and covariance of features from generated images, while $\mu_r$ and $\Sigma_r$ correspond to real images. Specifically, FID is calculated as:

$$\text{FID} = \|\mu_g - \mu_r\|^2 + \text{Tr}\left(\Sigma_g + \Sigma_r - 2(\Sigma_g \cdot \Sigma_r)^{1/2}\right). \tag{D.4}$$

Lower FID values indicate higher similarity between generated and real distributions, reflecting better image quality Ho et al. (2020); Song et al. (2021b); Dhariwal & Nichol (2021).
The *Number of Function Evaluations (NFE)* measures computational efficiency by counting neural network function calls during sampling Song et al. (2021b;a); Bao et al. (2022); Lu et al. (2022a); Karras et al. (2022). Lower NFE values indicate faster sampling.
Balancing FID and NFE is crucial for practical applications where both high-quality outputs and computational efficiency are required. Joint evaluation of these metrics provides a comprehensive perspective: FID assesses distribution fidelity, while NFE evaluates algorithmic efficiency.
In this paper, we adopt the evaluation framework used in prior studies Lu et al. (2022a;b), combining FID and NFE to jointly assess the quality of generated images and the computational efficiency of sampling algorithms. This comprehensive approach, validated in several studies Dhariwal & Nichol (2021); Song et al. (2021b); Lu et al. (2022a); Karras et al. (2022), offers a standardized benchmark for comparing different generative models and sampling methods.

### D.5 Conditional Sampling in Diffusion Models

Conditional sampling in DMs enables controlled generation by incorporating conditioning information (e.g., class labels or text) into the sampling process. This is achieved by modifying the noise predictor $\boldsymbol{\epsilon}_\theta(\boldsymbol{x}_t, t, c)$ to guide generation toward satisfying condition $c$. Two main approaches exist: *classifier-free guidance* Ho & Salimans (2021) and *classifier guidance* Dhariwal & Nichol (2021). Classifier-free guidance (CFG) combines conditional and unconditional predictions:

$$\boldsymbol{\epsilon}_\theta^{\text{CFG}}(\boldsymbol{x}_t, t, c) := (1 + w)\boldsymbol{\epsilon}_\theta(\boldsymbol{x}_t, t, c) - w\boldsymbol{\epsilon}_\theta(\boldsymbol{x}_t, t, \emptyset), \tag{D.5}$$

where $\emptyset$ denotes the unconditional case and $w > 0$ is the guidance scale. This method is simple and efficient as it requires no additional models.
Classifier guidance (CG) uses an auxiliary classifier $p_\phi(c \mid \boldsymbol{x}_t, t)$:

$$\boldsymbol{\epsilon}_\theta^{\text{CG}}(\boldsymbol{x}_t, t, c) := \boldsymbol{\epsilon}_\theta(\boldsymbol{x}_t, t) - s\sigma_t\nabla_{\boldsymbol{x}_t} \log p_\phi(c \mid \boldsymbol{x}_t, t), \tag{D.6}$$

where $s$ controls guidance strength and $\sigma_t$ is the noise level at time $t$. While computationally more expensive, this approach can provide finer control over the conditioning process.
In our experiments, we adopt the default guidance approach of the baseline method.

### D.6 Single-step Iteration Details

Our goal is to validate that variance-driven conditional entropy reduction can improve the denoising diffusion process. Compared to iterations based on traditional truncated Taylor expansions, RE-based iterations achieve better sampling performance, as demonstrated in DPM-Solver. This is because DPM-Solver iterations represent a specific instantiation of RE-based iterations, as shown in Proposition 3.3. Nevertheless, through extensive experiments on CIFAR-10 Krizhevsky (2009), CelebA 64 Liu et al. (2015), and ImageNet 256 Deng et al. (2009), we validated that RE-based iterations can further improve the denoising diffusion process by minimizing conditional variance. In this validation experiment, we adopt DPM-Solver Lu et al. (2022a) as our baseline. *Since the single-step iteration mechanism only requires the information from the starting point to the*

*information before the endpoint, RE-based iterations depend on prior variance assumptions to reduce the conditional variance between iterations.* Below, based on the principle of minimizing conditional variance, we demonstrate how to select parameters under the assumption of prior variance.

For clarity, we simplify the RE-based iteration in Eq. (4.1) as follows:

$$\boldsymbol{f}(\tilde{\boldsymbol{x}}_{t_{i-1}}) = \boldsymbol{f}(\tilde{\boldsymbol{x}}_{t_i}) + h_{t_i}\left(\left(\gamma_i + \frac{r_i}{2}\right)\boldsymbol{d}_\theta\left(\tilde{\boldsymbol{x}}_{s_i}, s_i\right) + \left(1 - \gamma_i - \frac{r_i}{2}\right)\boldsymbol{d}_\theta\left(\tilde{\boldsymbol{x}}_{t_i}, t_i\right)\right), \tag{D.7}$$

where $r_i = \frac{h_{t_i}}{\tilde{h}_{t_i}}$. To reduce variance of iteration (D.7) in each step, we configure the parameter $\gamma_i$ in accordance with the effective variance reduction interval prescribed in Proposition 4.1. Based on Proposition 4.1, since $\gamma_i \in \left[\frac{\text{SNR}(t_i)}{\text{SNR}(t_i)+\text{SNR}(s_i)}, \frac{\max\{2\cdot\text{SNR}(t_i),\ \text{SNR}(s_i)\}}{\text{SNR}(t_i)+\text{SNR}(s_i)}\right]$, when considering only $\gamma_i$ in isolation, we recommend three specific selections of prior parameter $\gamma_i$: $\gamma_i = \frac{\text{SNR}(t_i)}{\text{SNR}(t_i)+\text{SNR}(s_i)}$ and $\gamma_i = \frac{1}{2}$. Due to $r_i \in \left[1, \frac{4\ \text{SNR}(s_i)}{\text{SNR}(t_i)+\text{SNR}(s_i)}\right]$ based on Proposition 3.2. Based on the proof in B.1, when considering only $r_i$ in isolation, a ponential optimal value for $r_i$ is given by $\frac{2\ \text{SNR}(s_i)}{\text{SNR}(t_i)+\text{SNR}(s_i)}$. We recommend three specific selections of prior parameter $r_i$: $r_i = 1$ and $r_i = \sqrt{\frac{2\ \text{SNR}(s_i)}{\text{SNR}(t_i)+\text{SNR}(s_i)}}$. Based on empirical performance, we recommend the combinations $\left(r_i = 1, \gamma_i = \frac{1}{2}\right)$ or $\left(r_i = \sqrt{\frac{2\ \text{SNR}(s_i)}{\text{SNR}(t_i)+\text{SNR}(s_i)}}, \gamma_i = \frac{\text{SNR}(t_i)}{\text{SNR}(t_i)+\text{SNR}(s_i)}\right)$.

We compare the performance of RE-based iterations against several established solvers, including DDPM Ho et al. (2020), Analytic-DDPM Bao et al. (2022), DDIM Song et al. (2021a), DPM-Solver Lu et al. (2022a), F-PNDM Liu et al. (2022), and ERA-Solver Li et al. (2023). The comparative results are presented in Figures 2, 3, and Table 1. Remarkably, this consistent improvement in the conditional variance enhances image quality across various scenarios, as demonstrated by the ablation study with $\gamma_i = 1/2$ and $r_i = 1$ in Figures 2 and 3. Notably, compared to the 3.17 FID achieved by DDPM with 1000 NFEs Ho et al. (2020) on CIFAR-10, our RE-based iteration achieves a *3.15* FID with only *84* NFE, establishing a new *SOTA* FID for this discrete-time pre-trained model while realizing approximately *10×* acceleration. A visual comparison is shown in Figure 4.

### D.7 MULTI-STEP ITERATION DETAILS

In this section, we explore the potential of variance-based conditional entropy reduction to further enhance the denoising diffusion process. Unlike single-step mechanisms, multi-step iterations can leverage information from previous steps, providing additional context. Building on this advantage, we propose a training-free and efficient denoising iteration framework aimed at improving the denoising diffusion process through variance-driven conditional entropy reduction. Specifically, the framework minimizes conditional variance by reducing the discrepancies between actual states during the denoising iterations.

**Challenge.** Formulating the optimization objective to achieve this goal presents a significant challenge, requiring a mechanism that can effectively capture subtle state variations across iterations. The key lies in developing an algorithm that can identify meaningful features from state differences and transform these insights into signals that improve the denoising process. This involves not only quantifying state differences but also understanding the underlying deep information patterns in these variations, enabling more precise control over the denoising diffusion process.

**Conditional Variance Analysis.** Our analysis of the conditional variance reveals that the conditional variance in gradient-based denoising iterations can be composed of two critical components: the variance of the gradient estimation term itself and the variance between the gradient term and the first-order term. Specifically, revisiting the unified multi-step iteration in Eq. (4.7) induced in Section 4.2:

$$\boldsymbol{f}(\tilde{\boldsymbol{x}}_{t_{i-1}}) = \boldsymbol{f}(\tilde{\boldsymbol{x}}_{t_i}) + h_{t_i}\boldsymbol{d}_\theta\left(\tilde{\boldsymbol{x}}_{t_i}, t_i\right) + \frac{h_{t_i}^2}{2}\zeta_i\bar{B}_\theta(t_i; u_i). \tag{D.8}$$

This unified iteration can be rewritten as:

$$\boldsymbol{f}(\tilde{\boldsymbol{x}}_{t_{i-1}}) = \boldsymbol{f}(\tilde{\boldsymbol{x}}_{t_i}) + h_{t_i}\left(\left(1 \mp \frac{\zeta_i}{2}\frac{h_{t_i}}{h_{\mu_i}}\right)\boldsymbol{d}_\theta\left(\tilde{\boldsymbol{x}}_{t_i}, t_i\right) \pm \frac{\zeta_i}{2}\frac{h_{t_i}}{h_{\mu_i}}\boldsymbol{d}_\theta\left(\tilde{\boldsymbol{x}}_{\mu_i}, \mu_i\right)\right). \tag{D.9}$$

As $\boldsymbol{d}_\theta\left(\tilde{\boldsymbol{x}}_{\mu_i}, \mu_i\right)$ and $\boldsymbol{d}_\theta\left(\tilde{\boldsymbol{x}}_{t_i}, t_i\right)$ are known in multi-step iterative mechanisms, the $\text{Var}(\tilde{\boldsymbol{x}}_{t_{i-1}}|\tilde{\boldsymbol{x}}_{t_i})$ is determined by the value of $\zeta_i$, since $\frac{h_{t_i}}{h_{\mu_i}}$ is known. Inspired by Eq. (D.9), we observe that $\zeta_i$ balances the variance between the gradient term and the first-order term, optimally reducing conditional variance

by harmonizing their statistical characteristics. Therefore, we define $G(\zeta_i) = (1 - \zeta_i)\,\boldsymbol{d}_\theta\left(\tilde{\boldsymbol{x}}_{t_i}, t_i\right) + \zeta_i \boldsymbol{d}_\theta\left(\tilde{\boldsymbol{x}}_{\mu_i}, \mu_i\right)$ to balance the variance between these terms, as detailed in Section 4.3. Moreover, the variance of the gradient term itself requires careful balancing, as established in Proposition 4.2. Therefore, we introduced $\eta_i$ to balance the variance between the differences gradient, as detailed in Section 4.3.

**Optimization Objective and Algorithm.** Our optimization objective is formulated by considering both the discrepancy between actual data states and the variation in gradient states. Building upon these foundations, we outline this efficient conditional entropy reduction iteration mechanism driven by variance minimization in Algorithm 1, which offers an effective means to integrate variance-driven conditional entropy reduction into the denoising diffusion process by minimizing actual state differences.

**Practical Considerations.** Our goal is to develop an iterative denoising sampling algorithm for pre-trained diffusion models that requires neither additional training nor costly optimization procedures. However, in the iterative scheme aimed at minimizing the variance-driven conditional entropy reduction, we need to optimize the key parameters $\zeta_i$ and $\eta_i$ that control the conditional variance of the denoising iteration. As discussed in the main text, to balance optimality and computational efficiency, we adopt an optimization-guided streamlined approach to obtain optimized variance-reduction control parameters $\zeta_i$ and $\eta_i$. Specifically, the original optimization problem was a standard constrained mathematical programming problem. We observed that the problem possesses a closed-form solution when constraints are removed. Therefore, to directly obtain the optimized parameters in one step, we choose to apply an nonlinear non-negative mapping to this closed-form solution, using the mapped non-negative substitute as our final parameters. Since this non-negative substitute solution has already achieved the objective of quantifying the differences between states, it can serve as an effective alternative for parameter optimization, simultaneously ensuring computational efficiency and preserving the capability to capture critical state variations.

### D.7.1 Ablation Study

**Parameter settings.** In our implementation, we primarily employ the sigmoid activation function, which is one of the most prevalent activation functions in neural networks Rumelhart et al. (1986). Its mathematical expression is as follows:

$$\text{Sigmoid}(x) = \frac{1}{1 + e^{-x}}. \tag{D.10}$$

However, in our experiments, we found that the following improved version yields better results, particularly for high-dimensional datasets:

$$\zeta_i = \text{Sigmoid}\left(|\zeta_i^*| - \mu\right), \tag{D.11}$$

where $\zeta_i^*$ is computed using Eq. (4.16) and $\mu$ is a shift parameter introduced to fine-tune the solution space.

Conceptually, $\mu$ serves as a dynamic sensitivity regulator, allowing for nuanced control over the transformation of the activation function. By adjusting $\mu$, the inflection point of the sigmoid function can be shifted, effectively modulating the model's responsiveness to input variations across different regions of the input space. For high-dimensional datasets, this provides a principled mechanism for adaptive sensitivity calibration. The shift parameter enables more precise capturing of subtle state variations by expanding or contracting the function's most sensitive transformation region.

Empirical results show that this approach achieves a judicious balance between computational efficiency and the model's ability to discern critical state transitions. Table 5 systematically examines the impact of shift parameters on image generation performance in pre-trained diffusion models, using comprehensive ablation experiments on the ImageNet-256 dataset. Key findings include:

- *Global Performance Characteristics:* A consistent downward trend in FID scores is observed as NFE increases, indicating progressive refinement of sample quality. Performance differences among the tested shift parameters $\mu \in 0.25, 0.50, 0.75$ remain marginal, reflecting the *robustness* of the sampling process across configurations.

- *Shift Parameter Behavior Across NFE Stages:* Performance variations exhibit nuanced characteristics:

  - At lower NFE stages, performance differences between $\mu$ values are more pronounced.

- As NFE increases, the performance of different $\mu$ values converges.
- Different $\mu$ values exhibit unique progression patterns at various guidance scales, despite only marginal differences.

- *Impact of Guidance Scale:* The sensitivity to shift parameters varies with guidance scales:
  - Lower guidance scales (e.g., s=2) slightly more pronounced performance variations with $\mu$, with a change magnitude of 0.05 FID at 20 NFE.
  - As guidance scale increases (to s=3 and s=4), the influence of shift parameters becomes subtler and more stable, with a change magnitude of $0.02 \sim 0.03$ FID at 20 NFE.

Table 5: We conducted ablation experiments with different shift parameters in Algorithm 1, using the pre-trained model Dhariwal & Nichol (2021) on ImageNet-256 Deng et al. (2009). We report the FID ↓ evaluated on 10k samples for various NFEs and and guidance scales.

| Method | Guidance | Shift Parameter | NFE | | | | | | |
|--------|----------|-----------------|-----|-----|-----|-----|-----|-----|-----|
| | | | 5 | 6 | 8 | 10 | 12 | 15 | 20 |
| RE-2 | s=2 | $\mu = 0.25$ | **13.96** | **10.97** | 8.85 | 8.18 | 7.82 | 7.51 | 7.27 |
| | | $\mu = 0.50$ | 13.98 | 10.98 | 8.84 | 8.14 | **7.79** | **7.48** | **7.25** |
| | | $\mu = 0.75$ | 14.01 | 11.00 | **8.83** | **8.10** | 7.80 | 7.54 | 7.32 |
| RE-2 | s=3 | $\mu = 0.25$ | 14.43 | 11.08 | 8.90 | 8.30 | 7.92 | 7.58 | 7.53 |
| | | $\mu = 0.50$ | 14.37 | 11.04 | 8.87 | 8.31 | 7.89 | **7.56** | 7.51 |
| | | $\mu = 0.75$ | **14.32** | **10.99** | **8.85** | **8.23** | **7.87** | 7.56 | **7.50** |
| RE-2 | s=4 | $\mu = 0.25$ | 17.80 | 12.91 | 9.73 | 8.75 | 8.51 | **8.01** | 7.92 |
| | | $\mu = 0.50$ | 17.57 | 12.73 | 9.61 | 8.66 | **8.35** | 8.01 | 7.93 |
| | | $\mu = 0.75$ | **17.39** | **12.57** | **9.55** | **8.61** | 8.41 | 8.01 | 7.94 |

In summary, the properties of the aforementioned shift parameters collectively ensure the convergence and distinctiveness of our variance-driven conditional entropy reduction iterative scheme during the sampling process. Specifically, although these subtle variations are negligible on ImageNet-256, their distinctiveness is substantiated through experimental validation on stable diffusion model.

**Reducing Conditional Variance with Prior $r_i$.** In multi-step iterations, we require a probing step (an iteration step of Single-step Iteration Framework) to obtain the model value at the next state. Reducing conditional variance is crucial for improving the stability and accuracy of iterative algorithms, thus, we need to balance the conditional variance of the gradient term and the first-order term (see the above Conditional Variance Analysis part). We found that while logSNR typically performs well with larger step sizes, its advantages diminish as the NFE increases, as illustrated in Figure 2 and Table 6. For clarity, we revisit the logSNR as follows:

$$r_{\text{logSNR}}(t) = \frac{\log \frac{\alpha_t}{\sigma_t} - \log \frac{\alpha_{t+1}}{\sigma_{t+1}}}{\log \frac{\alpha_{t-1}}{\sigma_{t-1}} - \log \frac{\alpha_t}{\sigma_t}}. \tag{D.12}$$

This balance concept of logSNR leads to two potentially useful types of substitutions, which we present as follows.

From the perspective of balancing variances, one might consider the following form:

$$r_{\text{normvar}}(t) = \left( \frac{\text{Var}_{t+1} - \text{Var}_t}{\text{Var}_{t+1}} \right) \bigg/ \left( \frac{\text{Var}_t - \text{Var}_{t-1}}{\text{Var}_t} \right), \tag{D.13}$$

where $\text{Var}_t$ can represent any assumed variance, and satisfies $\text{Var}_t > \text{Var}_{t-1}$. If $\text{Var}_t < \text{Var}_{t-1}$, then simply swapping the roles of $\text{Var}_t$ and $\text{Var}_{t-1}$ in Eq. (D.13) will suffice.

Another substitution idea is to change the function space of the step size, for example, to the arctangent space:

$$r_{\text{arctan}}(t) = \frac{\arctan(h_t)}{\arctan(h_{t-1})}, \tag{D.14}$$

where $h_t$ denotes the step size from $t + 1$ to $t$.

In our experiments, we observed that a nonlinear combination of these two substitutions leads to improvements in certain scenarios. We define this nonlinear combination as *refined $r_i$*, and the ablation study of both logSNR and refined $r_i$ in the context of Algorithm 1 can be found in Table 6.

Table 6: We conducted ablation experiments under different guidance scales and different random seeds. Quantitative results of the gradient estimation-based denoising iterations using the pre-trained model Dhariwal & Nichol (2021) on ImageNet-256 Deng et al. (2009). We report the FID↓ for 10k samples evaluated under various numbers of function evaluations (NFEs). **Bold** values indicate the best FID in each iteration step column, while *italicized* values represent the second best.

| Method | Model | NFE | | | | | | |
|---|---|---|---|---|---|---|---|---|
| | | 5 | 6 | 8 | 10 | 12 | 15 | 20 |
| DPM-Solver++-2 | | 16.39 | 12.77 | 9.92 | 8.88 | 8.31 | 8.03 | 7.76 |
| DPM-Solver++-3 | | 15.64 | 11.64 | 9.21 | 8.51 | 8.12 | 7.97 | 7.69 |
| UniPC-2 | | 15.15 | 11.79 | 9.41 | 8.63 | 8.16 | 7.93 | 7.71 |
| UniPC-3 | Guided-Diffusion | 14.93 | 11.22 | 9.21 | 8.55 | 8.19 | 7.98 | 7.70 |
| DPM-Solver-v3-2 | (s=2, seed=1234) | 14.88 | *11.21* | 9.17 | 8.51 | 8.12 | 7.90 | 7.67 |
| DPM-Solver-v3-3 | | 15.62 | 11.73 | 9.57 | 8.89 | 8.37 | 8.01 | 7.65 |
| RE-2 ($r_i$=logSNR) | | **13.94** | **10.96** | 9.02 | *8.38* | *8.01* | *7.83* | *7.54* |
| RE-2 (refined $r_i$) | | *14.21* | *11.21* | *9.05* | **8.34** | **7.97** | **7.80** | **7.48** |
| DPM-Solver++-2 | | 16.62 | 12.86 | 9.73 | 8.68 | 8.17 | 7.80 | 7.51 |
| DPM-Solver++-3 | | 15.69 | 11.65 | 9.06 | 8.29 | 7.94 | 7.70 | 7.48 |
| UniPC-2 | | 15.37 | 11.78 | 9.22 | 8.40 | 8.01 | 7.71 | 7.47 |
| UniPC-3 | Guided-Diffusion | 15.05 | 11.30 | 9.07 | 8.36 | 8.01 | 7.72 | 7.47 |
| DPM-Solver-v3-2 | (s=2, seed=3407) | 14.92 | *11.13* | 8.98 | **8.14** | 7.93 | 7.70 | 7.42 |
| DPM-Solver-v3-3 | | 15.51 | 11.77 | 9.37 | 8.67 | 8.18 | 7.73 | 7.52 |
| RE-2 ($r_i$=logSNR) | | **13.98** | **10.98** | **8.84** | 8.16 | *7.81* | *7.52* | *7.32* |
| RE-2 (refined $r_i$) | | *14.33* | 11.16 | *8.95* | **8.14** | **7.79** | **7.48** | **7.25** |
| DPM-Solver++-2 | | 16.27 | 12.40 | 9.55 | 8.66 | 8.18 | 7.84 | 7.61 |
| DPM-Solver++-3 | | 15.93 | *11.49* | *8.98* | 8.39 | 8.11 | 7.74 | 7.63 |
| UniPC-2 | | *15.44* | 11.64 | 9.11 | 8.46 | 8.17 | 7.75 | 7.62 |
| UniPC-3 | Guided-Diffusion | 16.11 | 11.88 | 9.25 | 8.58 | 8.14 | 7.77 | 7.72 |
| DPM-Solver-v3-2 | (s=3, seed=3407) | 17.97 | 12.04 | 9.17 | 8.40 | 8.11 | 7.76 | 7.67 |
| DPM-Solver-v3-3 | | 20.87 | 14.94 | 10.68 | 9.29 | 8.57 | 7.92 | 7.77 |
| RE-2 ($r_i$=logSNR) | | **14.37** | **11.04** | **8.87** | *8.37* | **7.89** | **7.56** | 7.51 |
| RE-2 (refined $r_i$) | | 15.93 | 11.94 | 9.21 | **8.31** | **7.89** | *7.58* | *7.54* |

### D.7.2 COMPARATIVE ABLATION STUDY OF DENOISING SAMPLING METHODS.

To validate the effectiveness of Algorithm 1 in improving the denoising diffusion process, we conducted a comprehensive ablation study. We systematically compared various gradient estimation-based denoising sampling methods on ImageNet-256, a high-dimensional and complex dataset, using a baseline pre-trained model Dhariwal & Nichol (2021), employing multiple random seeds and varying guidance scales, as detailed in Table 6. To ensure fairness, the evaluation followed the standardized evaluation methodology outlined in D.4, with all methods were implemented using an identical codebase and default settings. Each method was implemented on the NVIDIA GeForce RTX 3090 GPU, utilizing a batch size of 25.

For the high-dimensional and complex ImageNet-256 dataset, advanced iterative methods such as DPM-Solver++ Lu et al. (2022b), UniPC Zhao et al. (2024), and DPM-Solver-v3 Zheng et al. (2023a) demonstrate steady performance improvements, progressively advancing the algorithm's capabilities. At NFE=5, the FID scores range from 16.62 (DPM-Solver++) to 14.88 (DPM-Solver-v3-2), and at NFE=20 , they converge within a narrower range of 7.42 to 7.76. Note that, among these three solvers, *DPM-Solver-v3 differs from DPM-Solver++ and UniPC* because it requires the extraction of additional empirical model statistics (EMS). To ensure clarity and fairness in our comparison, we categorize the three gradient-based sampling methods into two types for the purpose of discussing our improvements. Compared to DPM-Solver++ and UniPC, our proposed Algorithm 1 (RE-2 method), using both the log-SNR prior and the refined prior $r_i$ achieves consistently significant improvements, especially at lower NFEs. For example, at NFE=5, the log-SNR prior achieves an FID of **13.94**, while the refined prior $r_i$ reaches *14.21*, both clearly surpassing two gradient-based iterative methods,

which range from 16.62 to 14.88. At higher NFEs, both prior $r_i$ continues to demonstrate superiority, achieving *SOTA* performance with FID scores of **7.25** under the refined $r_i$, outperforming these methods (with scores ranging from **7.32** to **7.25**).

We observe that, compared to DPM-Solver++ and UniPC, DPM-Solver-v3 improves sampling performance on both high-dimensional and low-dimensional datasets by leveraging the concept of dataset-specific EMS. Notably, the improvement is particularly pronounced on low-dimensional datasets such as CIFAR-10 (see Table 3). Nevertheless, by applying the concept of variance-driven conditional entropy reduction iterative approach to the DPM-Solver-v3 framework, we also achieved improved performance, as presented in Table 3. Notably, on the pre-trained diffusion model on the ImageNet 256 dataset, our proposed RE-2 method *consistently improves sampling performance by leveraging variance-driven conditional entropy reduction iterations, with greater generalizability without requiring additional dataset-specific EMS*. For instance, while DPM-Solver-v3 achieved the best FID scores of 14.88 at NFE=5 and 7.42 at NFE=20, our method demonstrates improved FIDs of 13.98 at NFE=5 and 7.25 at NFE=20. Consequently, in contrast to DPM-Solver-v3, our algorithm not only improves the sampling performance but also holds greater generalizability, a characteristic particularly evident in our experiments on Stable Diffusion, as presented in Figure 7.

In summary, all ablation experiments on the general-purpose pre-trained model demonstrate that our proposed variance-driven conditional entropy reduction iterative framework *can consistently improve denoising diffusion sampling*. It achieves *SOTA* performance across different datasets and NFE levels, outperforming benchmark gradient-based sampling methods.

# E  SAMPLE COMPARISON IN PIXEL SPACE, LATENT SPACE, AND STABLE DIFFUSION

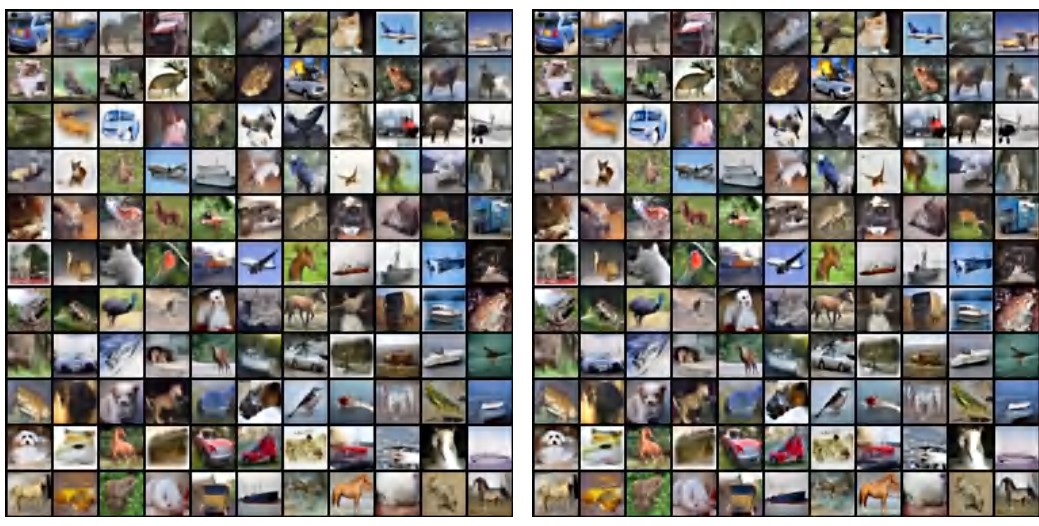

| (a) DPM-Solver-v3, 5 NFE , 12.22 FID. | (b) RE-based, 5 NFE, 10.61 FID. |

Figure 5: Random samples of EDM Karras et al. (2022) on CIFAR10 dataset with only 5 NFE. Within the EMS-parameterized iterative framework provided by DPM-Solver-v3 Zheng et al. (2023a), the RE-based iterative approach improves FID by balancing the conditional variance.

| DPM-Solver++ | UniPC | DPM-Solver-v3 | Algorithm 1 |
| 7.76 FID | 7.71 FID | 7.67 FID | 7.48 FID |

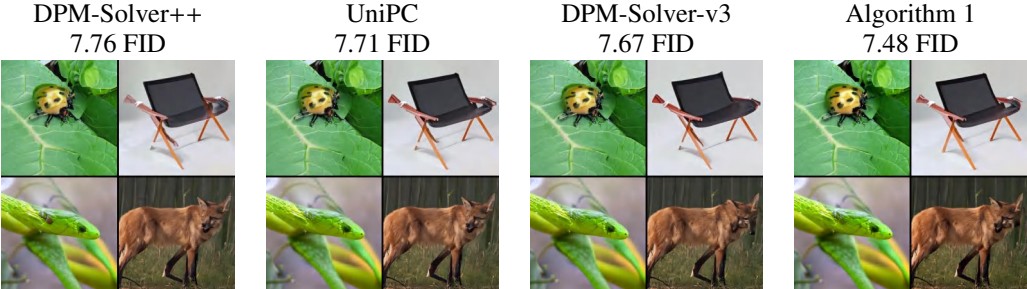

Figure 6: Random samples from the pretrained Guided-Diffusion model with 20 NFE on the ImageNet 256 dataset, Dhariwal & Nichol (2021).

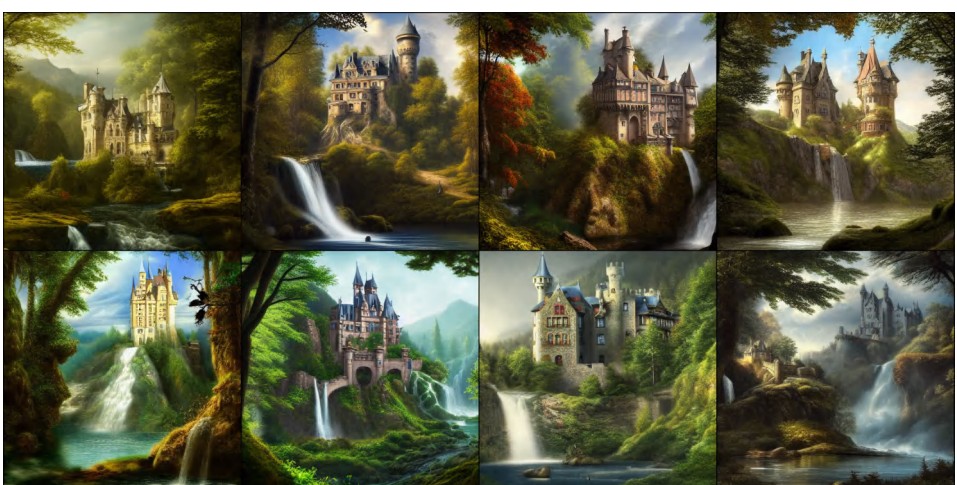

(a) Algorithm 1 (2m, s=0.75).

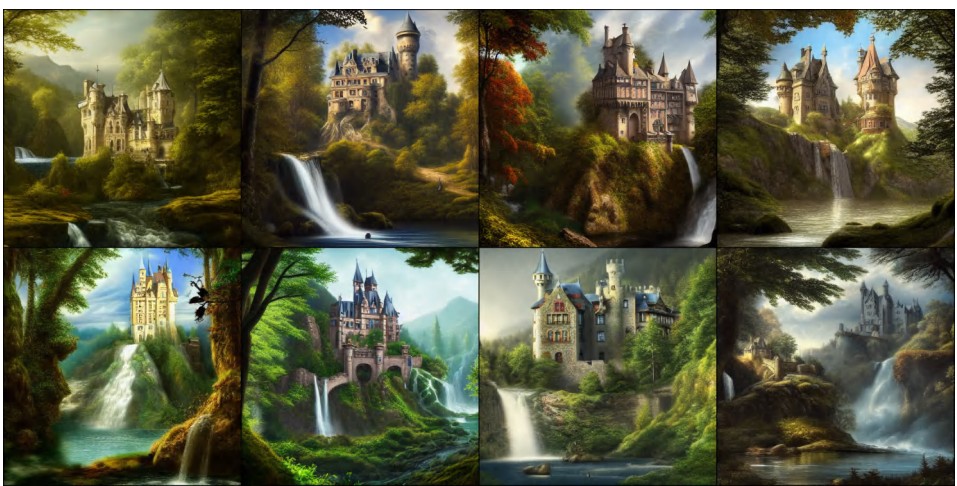

(b) Algorithm 1 (2m),s=0.5.

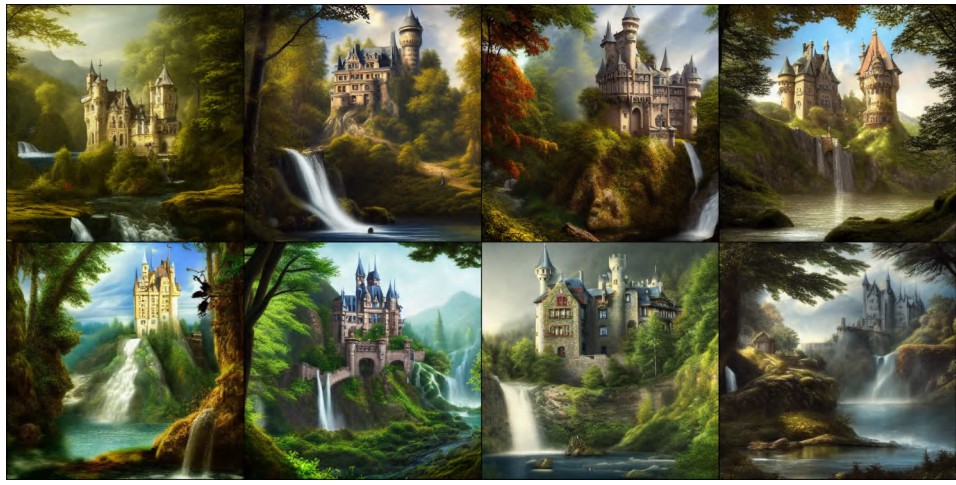

(c) Algorithm 1 (2m),s=0.1.

Figure 7: Random samples from Stable-Diffusion Rombach et al. (2022) with a classifier-free guidance scale 7.5, using 10 number of function evaluations (NFE) and text prompt "*A beautiful castle beside a waterfall in the woods, by Josef Thoma, matte painting, trending on artstation HQ*".

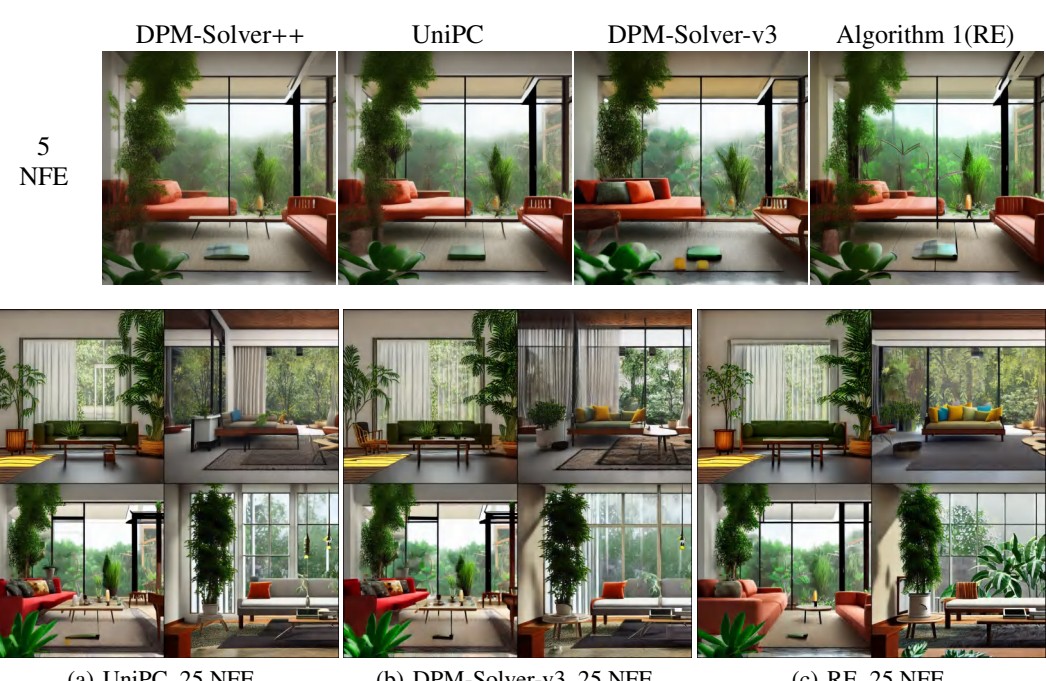

DPM-Solver++     UniPC     DPM-Solver-v3     Algorithm 1(RE)

5 NFE

(a) UniPC, 25 NFE.     (b) DPM-Solver-v3, 25 NFE.     (c) RE, 25 NFE.

Figure 8: Random samples from Stable-Diffusion Rombach et al. (2021) with a classifier-free guidance scale 7.5, with text prompt "*environment living room interior, mid century modern, indoor garden with fountain, retro, m vintage, designer furniture made of wood and plastic, concrete table, wood walls, indoor potted tree, large window, outdoor forest landscape, beautiful sunset, cinematic, concept art, sunstainable architecture, octane render, utopia, ethereal, cinematic light*".

NFE=5    NFE=10    NFE=15    NFE=25

DPM-Solver++ (Lu et al., 2022b)

UniPC (Zhao et al., 2024)

DPM-Solver-v3 (Zheng et al., 2023a)

Algorithm 1 (RE)

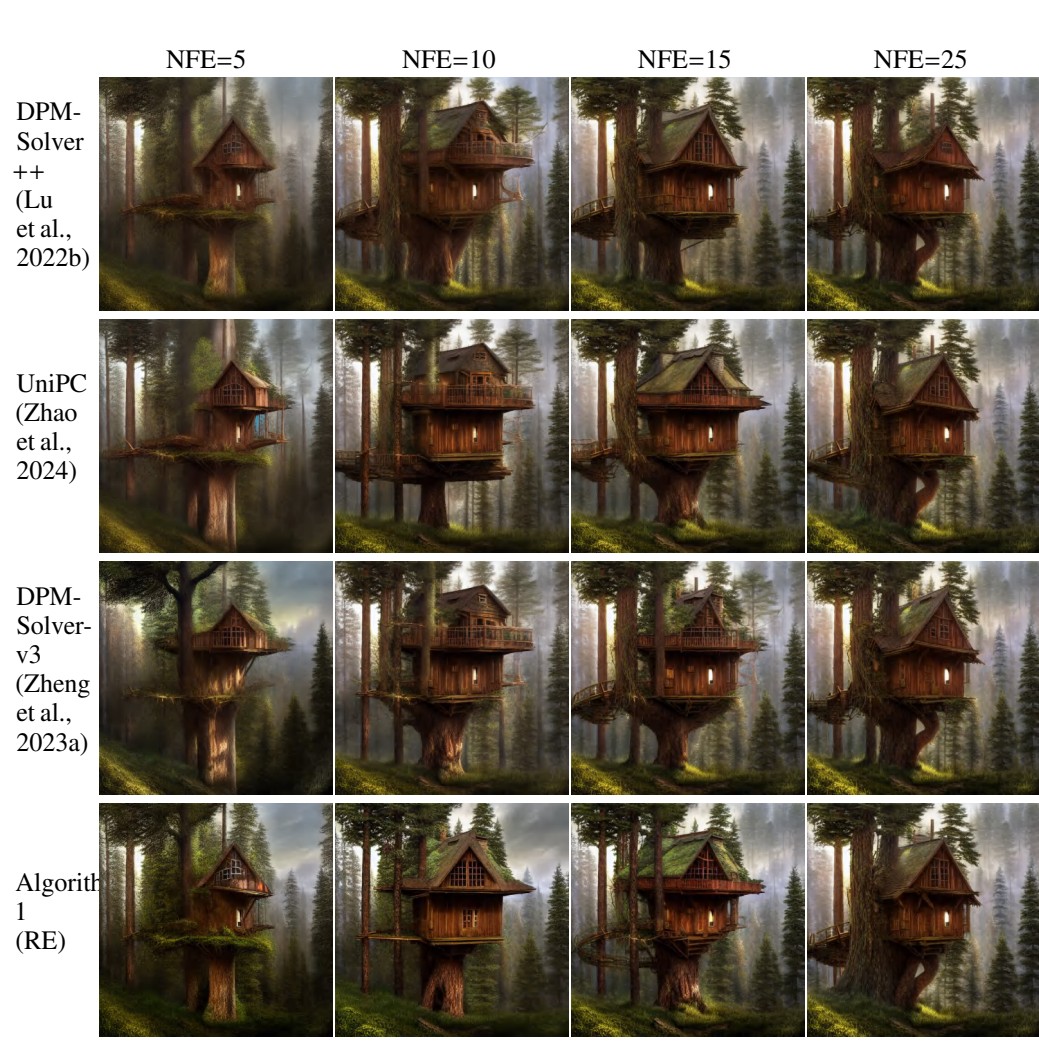

Figure 9: Random samples from Stable-Diffusion Rombach et al. (2021) with a classifier-free guidance scale 7.5, using 5, 10, 15, 25 NFEs and text prompt "*tree house in the forest, atmospheric, hyper realistic, epic composition, cinematic, landscape vista photography by Carr Clifton & Galen Rowell, 16K resolution, Landscape veduta photo by Dustin Lefevre & tdraw, detailed landscape painting by Ivan Shishkin, DeviantArt, Flickr, rendered in Enscape, Miyazaki, Nausicaa Ghibli, Breath of The Wild, 4k detailed post processing, artstation, unreal engine*".

DPM-Solver++ (Lu et al., 2022b)

UniPC Zhao et al. (2024)

DPM-Solver-v3 Zheng et al. (2023a)

Algorithm 1

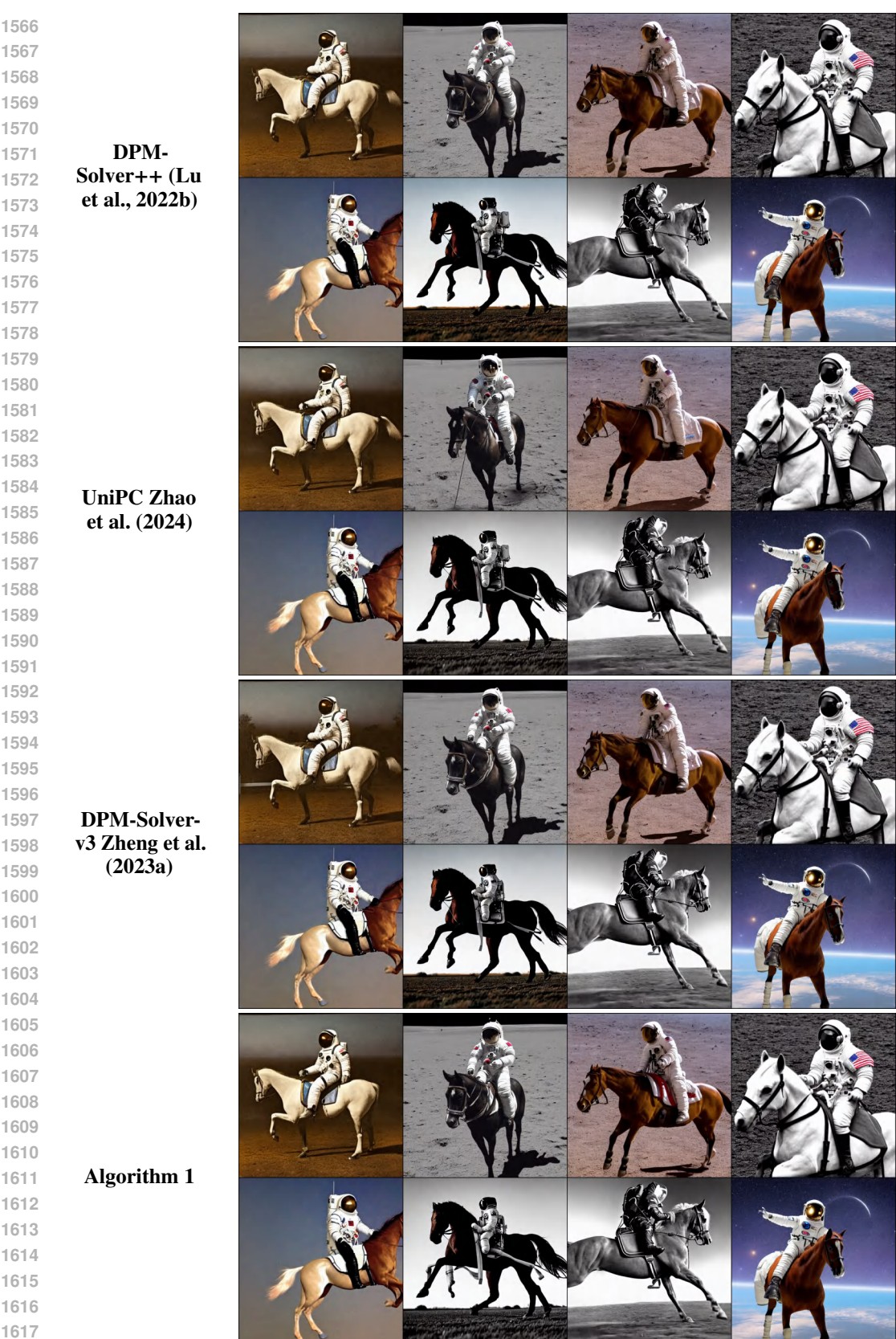

Figure 10: Random samples from Stable-Diffusion Rombach et al. (2022) with a classifier-free guidance scale 7.5, using 50 NFE and text prompt "*a photograph of an astronaut riding a horse*".

