# OpenReview forum: "Improving Denoising Diffusion with Efficient Conditional Entropy Reduction"
_ICLR.cc/2025/Conference — Submitted to ICLR 2025_

### Official Review · Reviewer_dkF9 · 2024-10-26

**Soundness:** 2
**Presentation:** 2
**Contribution:** 2
**Rating:** 5
**Confidence:** 1

**Summary:**

This paper examines the efficiency of denoising diffusion models (DMs) by introducing an entropy-focused approach to improve denoising iterations. It proposes conditional entropy reduction as a means to streamline denoising iterations, leveraging gradient estimation techniques without additional model training. This method aims to enhance the denoising efficiency, theoretically and empirically validating it on pre-trained models, thereby promising a more optimized and effective denoising process. The paper’s contributions include an analysis of entropy's role in denoising, proposing improved iterations for entropy reduction and empirically confirming the benefits of this method on image and text-to-image models.

**Strengths:**

This paper proposes an innovative approach to enhancing denoising diffusion models (DMs) by introducing **conditional entropy reduction** as a framework for improving denoising efficiency. This concept shifts the focus from purely model-centric improvements to a novel mathematical approach, treating entropy as a direct metric for accelerating diffusion processes. By employing **RE-based (entropy reduction-based) iterations**, the authors effectively reduce computational demands and improve output quality. This approach creatively combines elements from information theory and generative modeling, marking a significant departure from traditional DM acceleration techniques. Such application of entropy, directly aimed at process optimization within diffusion models, demonstrates originality and has not been explicitly explored in prior works.

The paper's quality is evident in its rigorous theoretical framework, well-documented experimental design, and thorough comparative analysis. The authors carefully derive the principles of entropy reduction in diffusion models, offering clear justifications for each proposed method, including the **gradient estimation-based iteration** to achieve conditional entropy reduction. Theoretical insights are validated across diverse benchmarks, such as **CIFAR-10** and **ImageNet**, showcasing improvements in **Fréchet Inception Distance (FID)** scores. The comparisons with established methods, including **DDIM** and **DPM-Solver**, reflect a robust and comprehensive experimental framework. By testing across both **single-step and multi-step scenarios**, the paper demonstrates the practical reliability and consistency of its proposed methods, underscoring a strong commitment to research quality.

The clarity of the paper is generally commendable, particularly in its explanations of complex mathematical derivations and the entropy reduction approach applied to the denoising process. Key concepts, including **conditional entropy, score-integral estimation, and gradient-based iterative improvements**, are introduced systematically, making them accessible to readers familiar with diffusion models. Visual aids, such as tables and diagrams comparing FID scores across conditions, effectively illustrate performance improvements, enhancing readability. However, the technical depth, especially in sections like the proofs of **Propositions 3.1 and 3.2**, may present a challenge to those without a specialized background. Despite this, the structured presentation helps readers grasp the main contributions of the proposed methods to the denoising process.

The paper’s significance is substantial, addressing critical limitations of current diffusion models, notably their high computational costs and the iterative inefficiency in denoising. By reducing conditional entropy at each step, the paper achieves enhanced denoising quality and efficiency, while also providing a valuable entropy-centric framework that could apply to other generative modeling tasks. This entropy reduction perspective provides a foundational shift that could potentially expand to broader applications requiring high-quality generative outputs in fields like **image and video synthesis, text-to-image generation, and voice synthesis**. The methodology outlined here holds promise for influencing future model design by establishing entropy as a target for iterative refinement, potentially impacting other AI subfields that prioritize both efficiency and quality in generative modeling.

In conclusion, this paper demonstrates a high degree of originality through its entropy-focused approach, maintains rigorous research standards, communicates complex methodologies with considerable clarity, and offers significant potential for advancing the field of diffusion models and generative modeling. By introducing conditional entropy reduction as a tool for accelerating and enhancing denoising diffusion, the paper presents a compelling framework with wide-ranging implications for applications demanding efficient and high-fidelity denoising processes.

**Weaknesses:**

This paper offers a robust theoretical framework with innovative contributions to diffusion models (DMs) through the use of conditional entropy reduction for enhanced denoising efficiency. However, there are several areas where improvements could significantly increase the paper's accessibility, generalizability, and practical utility.

Firstly, while the paper provides detailed theoretical support, certain sections, particularly the proofs for **Propositions 3.1 and 3.2** and the derivation of gradient estimation-based iterations, are dense and technically complex. This high level of mathematical rigor may make the work less accessible to readers who lack a specialized background in advanced calculus or information theory. Simplifying or supplementing these proofs with intuitive explanations and illustrative diagrams could make the core ideas more accessible, potentially widening the audience and facilitating broader adoption of the methodology.

Secondly, the empirical validation, though robust on image-based benchmarks like **CIFAR-10** and **ImageNet**, lacks diversity across different types of generative tasks and data. The experiments focus primarily on image data, which limits the conclusions that can be drawn about the method’s effectiveness in other areas where DMs are applied, such as **text-to-image, audio, or video generation**. Extending the validation to these domains would provide stronger evidence of the proposed method’s versatility and demonstrate its broader applicability to different generative modeling challenges.

Furthermore, the concept of **entropy reduction** is central to the proposed approach, but the paper could further elucidate how this metric directly influences model performance at each iterative step. Expanding on the relationship between conditional entropy reduction and the quality of denoising could offer readers a clearer understanding of the metric's impact on performance. Including visualizations that show the changes in entropy and their correlation with output quality at each iteration could deepen insights into this relationship and enhance the clarity of the work.

Another practical limitation lies in the adaptability of **RE-based iterations**. Researchers and practitioners who aim to incorporate these iterations into diverse architectures or applications may face challenges, especially if existing infrastructures require substantial adjustments for entropy reduction integration. More explicit guidance on adapting these iterations for different DM variants would improve the paper’s practical utility. Adding modular guidelines or pseudocode, as well as detailing compatibility with standard DM frameworks, would facilitate adaptation and encourage further experimentation in real-world applications.

In summary, this paper would benefit from more accessible explanations of its mathematical rigor, broader empirical validation across diverse generative tasks, a clearer discussion of the role of entropy in denoising, and practical guidance on integrating RE-based iterations into different DM architectures. Addressing these areas could significantly broaden the appeal and applicability of the work, allowing it to better fulfill its potential as a foundational advancement in efficient, high-quality diffusion modeling.

**Questions:**

-  Could you provide a more detailed explanation of how conditional entropy reduction directly influences the quality of denoising at each iterative step? Specifically, how does lower conditional entropy correlate with higher-quality generative outputs in a measurable way?



-  How adaptable are the RE-based iterations to different types of diffusion models, such as those used in text-to-image generation or audio synthesis? Are there any known limitations or additional requirements when applying RE-based iterations to these variations?


 - Do you have plans to test RE-based iterations on tasks beyond image synthesis, such as text-to-image, audio, or video generation? If not, could you discuss any anticipated challenges in applying the method to these domains?

-  The mathematical proofs, especially for **Propositions 3.1 and 3.2**, require a strong background in advanced calculus and information theory. Could you provide more intuitive explanations or visual aids to support readers who may find these sections challenging?


 - Have you measured the computational overhead introduced by the RE-based iterations? How does the entropy reduction approach impact memory usage, runtime, and scalability for larger datasets or more complex models?


- For researchers interested in implementing RE-based iterations in their diffusion models, could you provide pseudocode or a high-level description of the integration steps?



-  Did you observe any trade-offs between denoising speed and output quality when using RE-based iterations? If so, how might these trade-offs vary across different types of models or data?



- Have you established any baseline or threshold levels for conditional entropy across typical datasets (e.g., CIFAR-10, ImageNet) that could serve as reference points for evaluating model performance?

- Beyond FID scores, have you evaluated RE-based iterations on other metrics (e.g., perceptual quality, accuracy of specific content features)? If not, are there particular metrics that may benefit from future analysis?

**Details Of Ethics Concerns:**

There are no ethical concerns with this research.

---

> ### Author Response · Authors · 2024-11-30
>
> Dear Reviewer,
>
> Thank you for your meticulous  feedback.  Below are our specific responses:
>
> *1. Theoretical Complexity and Readability*
>
> We've added intuitive explanations to improve manuscript readability.
>
> Subtitles for Clarity:
> - In Section 3.2, we have added subtitles at the beginning of key subsections to summarize their content succinctly. We have also elaborated on the relationships between different concepts and provided explanations of the assumptions, including their purposes and limitations, to help readers better contextualize our work. We have highlighted the main revisions using color coding.
> - Relocating Proofs to the Appendix:
> To make our manuscript more accessible, we have moved the detailed proofs in Section 3.2 to the appendix. References to the proofs are provided at key points in the text for readers who wish to examine them.
> - Enhancing Intuitive Understanding:
> We have included more intuitive explanations of the propositions where appropriate to lower the barrier to understanding.
> - Improved Organization of Key Content:
> To aid readers in better grasping the content of Propositions 3.1 and 3.2, we have introduced a key statement of  *Variance-Driven Conditional Entropy Reduction in Gradient-Based Iterations*. This title provides a clear structure, helping readers quickly locate and comprehend the foundation of these propositions.
> - Explanations of Core Concepts:
> We have supplemented the main text with brief explanations of key concepts such as conditional entropy and gradient estimation, making the paper more accessible to readers with diverse backgrounds.
>
> *2.Correlation between conditional entropy and denoising quality*
>
> From an information-theoretic perspective, conditional entropy provides a fundamental measure of denoising performance. Low conditional entropy signifies effective noise removal, as it indicates a more concentrated probability distribution of the output image when a denoising model successfully reconstructs the original image from a noisy input.
> Specifically, an efficient denoising algorithm produces an output with low conditional entropy, meaning minimal uncertainty given the noisy input. Conversely, when the algorithm fails to effectively remove noise, the output image exhibits higher conditional entropy, reflecting greater uncertainty and lower image recovery accuracy.
> In diffusion models, each reverse conditional transition distribution represents the probability  $p(x_t | x_{t+1}, x_0)$. Under the Gaussian assumption, the conditional entropy is fundamentally determined by the conditional variance. Theoretically, a smaller conditional variance implies that $x_t$ is closer to both $x_{t+1}$ and $x_0$, which directly correlates with improved denoising quality at each iterative step.
>
> Regarding evaluation metrics, for fairness, we follow the standard FID and NFE metrics used by previous works to measure sampling performance, more specialized metrics will be worth exploring in the future. We have detailed these metrics  in Appendix D4.
>
> *3. Diversity of Experimental Validation and Adaptability of the Method*
> Thank you for pointing out the limitations in our experimental validation. Although our original manuscript already included comparative experiments on CelebA, in response to your suggestion, we have expanded our method's comparative experiments on Stable Diffusion and provided comparisons with  the most advanced training-free sampling methods in text-to-image generation in the main text, with more detailed comparisons available in Appendix E. Furthermore, we conducted extensive ablation studies on ImageNet-256.
> Our research demonstrates that the variance-driven conditional entropy reduction iteration exhibits robustness across parameters and validates our method's consistent superiority over  the most advanced  sampling methods under various conditions. This proves that our proposed approach not only has theoretical potential for achieving superior performance but also attains SOTA sampling performance in practical comparisons, consistently improving training-free sampling methods for diffusion models. Additionally, our paper verifies that the variance-driven conditional entropy reduction iteration consistently improves sampling performance from different parameterization perspectives, such as noise-based, data-based, and DPM-Solver-v3-based approaches.
>
> *4. Computational Cost and Efficiency Analysis*
>
> The iterative approach based on RE-based sampling we propose is a purely training-free method, meaning that no additional high-cost optimization or training overhead is introduced. The RE algorithm is a standard iterative method, where the parameter introduced to reduce conditional variance is obtained in a single step, without incurring any additional optimization costs. Overall, the computational efficiency of our proposed method is on par with the DPM-Solver++ benchmark.
>
> Once again, we sincerely appreciate your valuable feedback.

---

### Official Review · Reviewer_XV1u · 2024-10-28

**Soundness:** 2
**Presentation:** 2
**Contribution:** 3
**Rating:** 5
**Confidence:** 4

**Summary:**

This paper finds that gradient estimationbased iterations enhance the denoising process by effectively reducing the conditional entropy of reverse transition distribution. Therefore, this paper introduces streamlined denoising iterations for DMs that optimize conditional entropy in score-integral estimation to improve the denoising iterations. The effectiveness of this method is verified on both pixel and latent spaces diffusion models.

**Strengths:**

1. The concept that the condition entropy and the inference acceleration are correlated is interesting.
2. The theoretical derivation is sufficient.
3. Superior performance over the previous samplers on the pixal and latent space diffusin models.

**Weaknesses:**

1. The writing of this paper is poor. Certain key concepts are not defined or defined at latter page, for example, the concept of conditional entropy. Too much theoretical derivation and proof hinders the better expression of this paper as well as the understanding of readers. It is suggested to leave proof in the appendix, and highlights the crucial theroical findings and conclusions.
2. The intorduction of conditional entropy is abrupt. Uncertainty reduction seems more like the result of inference convergence, instead of the inner reason. This also applies to the concept of conditional entropy. Compared to x_{t+1}, x_t has lower noise level. It is thus also natural that x_t has less uncertainty than x_{t+1}.
3. Among all the proposed assumptions, how to detailly inplement the proposed method seems uncler and less strengthed.
4. The visual results are embarrassedly absent. Besides the quantitative results, the visual results are also highly required.
5. For the only proposed visual results, it is found that the proposed method may destroy the determinestic nature of ODE sampler. For example, compared to DDIM, the proposed method has different structures.

**Questions:**

Refer to the weakness part. It is highly suggest to improve the fluency and cores of this paper. More visual results are also encouraged.

---

> ### Author Response · Authors · 2024-11-29
>
> Dear Reviewer,
>
> We  sincerely thank you for your constructive feedback. Below are our detailed responses to the points you raised:
>
> *1. Improving clarity*
>
> We have enhanced the clarity of Sec. 3.2. Specifically, we added  subtitles to summarize each major topic, clarified several definitions and relationships, and utilized the freed-up space to improve the  completeness of the experimental descriptions, with more detailed   provided in Appendix (App.) D. Following your suggestion, we moved a proof from Sec. 3.2 to App. B1 and used the additional space to include pseudocode for variance-driven conditional entropy reduction iterations, as shown in Algorithm 1 on page 8. Further, we added the conclusion to better highlight our contributions.
>
> *2. Conditional entropy*
>
> To introduce conditional entropy more clearly, we have refined the explanation of its connection with denoising iterations in as much detail as possible. Under the rules of denoising iterations, the reduction of conditional entropy aligns with the goal of minimizing reconstruction error. Specifically, under the denoising framework, the direction of efficient conditional entropy reduction corresponds to the direction of minimizing reconstruction error.
>
> *3.  Method implementation*
>
> We present two approaches:
>
> Prior-Based Approach. By leveraging the known noise levels of the model at each time step, we utilize the prior variance of these noise levels to enhance traditional gradient-based iterations.   As demonstrated in our work,  the DPM-Solver iterations distinguish themselves from conventional methods by employing logSNR to balance the conditional variance during iterations.   Our approach focuses primarily on reducing or balancing the conditional variance of the current iteration mechanism, while being largely indifferent to the specific space in which the Taylor approximation is performed. In page 26, we provide two alternative forms for reducing conditional variance (Eq. (D.13) and Eq. (D.14)), and our empirical findings show these methods outperform traditional gradient-based iterations. Additionally, we offer an implementation of this prior-based approach in App. D6.
>
> Non-Prior Approach (*Core Contribution*). Our second approach represents a more substantial improvement over prior-based methods and constitutes the core practical contribution of our work. We  overcame the **challenge** of formulating an optimization objective that liberates the mechanism from prior assumption cases. Specifically, we *identified and captured* the key parameters that control the reduction of conditional variance. Then, we developed an optimization objective for these critical parameters aimed at minimizing conditional variance. As detailed in Section 4.3, we improved these key parameters by: Minimizing the difference in  states; Balancing differential approximations of gradient. We determined the parameter $\zeta_i$ by minimizing the difference between the approximated states $x_{t1}$ and $x_{t2}$ (obtained from forward and reverse processes) and the true state at time t. Fundamentally, this minimization reduces the conditional entropy relative to the true state and clean data.
> Simultaneously, minimizing the difference between the estimated and true gradients essentially reduces the error in gradient estimation. More detailed content can be found in App. D7 and Sec. 4.3.
> Given that conventional optimization algorithms typically require multiple iterative refinements, which can burden sampling efficiency, we transform the closed-form solution available in special cases to directly substitute the final parameters, thereby avoiding additional optimization overhead. Our experimental results validate the consistent improvements of this approach. Compared to baseline methods like DPM-Solver++ and other advanced solvers, our variance-driven, conditional entropy reduction iteration achieves SOTA performance across various scenarios on high-dimensional datasets such as ImageNet256.
>
> *4-5. Visual comparisons and deterministic nature.*
>
> We sincerely thank the reviewer for their encouraging feedback, which has motivated us to further refine this work. We have added more visual comparisons in Figure 1 and App. E to address the reviewer's concern. Regarding the observation that our method may differ from DDIM, we acknowledge that such differences are expected, as our method introduces variance-driven conditional entropy reduction iterations.  While these differences may lead to structural variations, the generated images remain deterministic and high-quality, as shown by the quantitative results in Figs. 9 and 10. We view this methodological diversity as a potential advantage, similar to how different solvers may deviate from DDIM in various ways. Perhaps the key consideration is not just the diversity these solvers introduce, but their stability moving forward.
>
> We hope these clarifications address your concerns.  Thank you once again.

---

### Official Review · Reviewer_tEg4 · 2024-11-02

**Soundness:** 1
**Presentation:** 1
**Contribution:** 2
**Rating:** 3
**Confidence:** 3

**Summary:**

The paper proposed two methods for solving the ODE formulation of the reversed diffusion flow.
The methods are motivated by claims on reduction of conditional entropy.

**Strengths:**

* The proposed iteration scheme seems to be empirically useful when comparing with existing ODE-based methods.

**Weaknesses:**

- The presentation is not good enough.
Clearly, it also lacks discussion in the experiments section and conclusion.

- Many claims do not seem to be well justified and some formal statements are not written in a clear and explicit way.

- Most of the results are based on assumption in the reverse flow that is not justified.
Therefore, the proposed methods are not really mathematically backed.

**Questions:**

- I tend to believe that in the *reversed flow* you cannot assume that x_{t} conditioned on x_{t+1} is Gaussian (with expectation x_{t+1}).
Note that conditioned on both x_{t+1} and x_{0}, we have that x_{t} is Gaussian (with suitable expectation and variance), but not when dropping the conditioning on x_{0}.
This makes Eq 3.9 (using the formula of entropy of Gaussian RV) unjustified.

- Furthermore, in practice, the reversed flow is conducted by discretization and integration, which means that analyzing it by properties of the continuous ODE are not rigorous.

- The following sentence seems like hand-waiving:
"Since the injected noise at different time steps in a DM is mutually independent, the estimated noise by the model at different time steps can also be regarded as mutually independent."
In the sequential denoising operations of the solvers I tend to believe that you will see correlation.

---

> ### Author Response · Authors · 2024-11-29
>
> Dear Reviewer,
>
> Thank you for your comments. Below are our detailed responses to the points you raised:
>
> *Presentation and Clarity*
>
> In our revised manuscript, we have made the following efforts:
> - We have revised the manuscript for better readability and clarity. Such as,  we provided a subtitle at the beginning of some core content in Sec. 3.2  to appropriately summarize its content and highlighted our revisions using color coding for easy identification.
> - We have moved some proofs to Appendix. With the space saved and in line with ICLR's new 10-page limit, we have expanded the discussion in the experimental section,  with more detailed content provided in the added Appendix D. We added the paper's conclusion to improve the clarity of our research contribution.
>
> *Gaussian Assumption*
>
> - We have clarified that $x_{t_i}$ is conditioned on both $x_{t_{i+1}}$ and $x_0$,   and explained that  $x_{t_i}|x_{t_{i+1}}$ is shorthand for this conditional relationship, as seen in page 4.   In fact, our earlier version was driven by this assumption, as used in our proof in the appendix.
>
> - Based on Eq. (3.10),  we have modified the description of "conditional entropy reduction" to "variance-driven conditional entropy reduction" in main text to better clarify our contribution.
>
> *The reversed flow ODE and Discretization*
>
> We agree with your observation that analyzing the reversed flow using continuous ODE properties in a discretized setting is not rigorous.  Clearly, the primary focus of this paper is **not on the properties of the ODE in its continuous form**.  *Instead, we focus on the denoising iterative errors introduced by the discretization process*. We only state that ODEs provide a principled mechanism.  Our analysis centers on the gradient-based denoising process.  Our motivation  is that gradient-based iterations suffer from errors.  In fact, our viewpoint is consistent. Our core contribution is to reduce these iterative errors by lowering the conditional variance, which improves the denoising process. Our experiments validate the proposed solution.
>
> *Independence Assumption in the Reversed Flow*
>
> Thank you for pointing out the limitations of the noise independence assumption. In our revisions, we have clarified that  the neural network in the model shares parameters across different timesteps, which may introduce correlations in the predicted noise.
> However, relevant works, such as Song et al. [1], analyze the theoretical gap between models that assume independent noise and those that account for parameter-sharing-induced correlations. Their work justifies the use of the independence assumption as a reasonable approximation, showing that while there may be some performance difference, the impact of this simplification is generally small. This suggests that the independence assumption can still be a useful surrogate for theoretical analysis, especially given its alignment with training objectives like the mean squared error loss. To prevent any potential misunderstandings, we have explicitly stated the limitations of this assumption in the revised manuscript.
>
>
> We are truly grateful for your  comments, which have  contributed to improving the clarity and rigor of our work. We believe the revisions, particularly regarding the noise independence assumption and the theoretical considerations, have strengthened the manuscript and addressed the concerns you raised. We sincerely hope that the changes we have made are now clearer and more convincing, and we deeply appreciate the time and effort you have invested in reviewing our paper.
>
> [1]. Song et al., Denoising diffusion implicit models, ICLR 2021.

---

### Official Review · Reviewer_PjCg · 2024-11-04

**Soundness:** 3
**Presentation:** 3
**Contribution:** 3
**Rating:** 6
**Confidence:** 4

**Summary:**

This paper presents an interesting approach for reducing entropy during reverse sampling.  It theoretically derived the coefficients for sampling that reduce the conditional entropy the most. Results showing improvement in FID over samplers like DPM++.

**Strengths:**

1. Well written
2. Clear to understand
3. Mathematical sound
4. Interesting perspective on improving sampling.

**Weaknesses:**

1. The motivation of reducing entropy may need elaboration. Why this necessarily improves sampling quality?
2. The result shows significant degradation on FID when NFE reduces from 20 to 5, which casts doubt on how this method can actually helps with few-step sampling. This paper needs to compare with other distillation methods such as CM [1] or LCM [2].
3. More results with different CFG should be desirable. It is known that higher CFG may cause instability in sampling. How is the FID performing at different CFG scales



----------------------------------------------Post Rebuttal------------------------------------------------------------------------------------------


The authors address my questions well. I think this is an interesting and solid paper but need more theoretical analysis on the conditional entropy. Also the improvement of performance at fewer than 10 NFEs is still challenging. I maintain my original rating.



[1] Song, Y., Dhariwal, P., Chen, M., & Sutskever, I. (2023, July). Consistency Models. In International Conference on Machine Learning (pp. 32211-32252). PMLR.

[2] Luo, Simian, et al. "Latent consistency models: Synthesizing high-resolution images with few-step inference." arXiv preprint arXiv:2310.04378 (2023).

**Questions:**

1. The motivation of reducing entropy may need elaboration. Why this necessarily improves sampling quality?
2. The result shows significant degradation on FID when NFE reduces from 20 to 5, which casts doubt on how this method can actually helps with few-step sampling. This paper needs to compare with other distillation methods such as CM [1] or LCM [2].
3. More results with different CFG should be desirable. It is known that higher CFG may cause instability in sampling. How is the FID performing at different CFG scales
4. Can this method be applied to image editing tasks with inversion?

---

> ### Author Response · Authors · 2024-11-29
>
> Dear Reviewer,
>
> We would like to sincerely thank you for your constructive feedback and positive support for our work, which has been truly inspiring to us. Below are our detailed responses to the points you raised:
>
> *1.The motivation for reducing entropy*
>
> Under the Gaussian assumption framework, reducing conditional entropy is equivalent to reducing conditional variance. In diffusion models, since the conditional distribution is conditioned on the clean data, reducing the conditional variance implies reducing the distance from the clean data. This suggests that reducing conditional variance has the potential to improve the sampling process. Moreover, based on our analysis and observations, the DPM-Solver iteration, which differs from traditional iterations based on Taylor series truncation, essentially represents an iteration that reduces conditional variance. This is why DPM-Solver demonstrates superior performance compared to traditional gradient-based iterations. Motivated by these theoretical insights and empirical observations, our research is driven by the question of whether there exists a more efficient way to reduce conditional entropy during the iterative process. As observed in our experiments, our variance-driven conditional entropy reduction iteration *differs from* other iterative methods. Interestingly, it (the RE iteration) appears to be dedicated to ensuring the *completeness*, *standardization*, and *clarity* of objectives at *each NFE stage*, as illustrated in Figure 9 of the appendix.
>
> *2. This paper needs to compare with other distillation methods such as CM [1] or LCM [2]*
>
> We greatly appreciate your concern, which provides us an opportunity to clarify our work. In response to your comment, we have made an effort to briefly summarize the development of diffusion models and their efficient variants in Appendix A (under the "Related Work" section). While this may not be exhaustive, it is sufficient to clarify the positioning of our work. In general, CM and LCM are efficient models built on top of diffusion models, relying on training or re-training processes. Typically, such training-based efficient models require prior knowledge, often provided by the original diffusion model acting as a "teacher." This prior knowledge is then obtained by sampling using  a training-free sampling method based on diffusion models. In contrast, our work focuses specifically on the training-free sampling methods in diffusion models, as stated in the abstract. Given the significant differences in cost and mechanisms between these two types, a direct comparison of these sampling methods would not be fair. Future research might explore applying our insights to knowledge distillation.
>
> *3.More results with different CFG should be desirable.*
>
> We truly value your professional suggestion. In Appendix, Tables 5 and 6, we have conducted ablation experiments under different CFG settings. Compared to the most advanced methods in the same category, our approach consistently shows improvements across various scenarios. We also performed ablation on the displacement parameters within our iterative method to validate its robustness.
> Additionally, we list the best results obtained on our server for each method when  $s=2$ in the CFG   as follows:
>
> | Guided-Diffusion256      | 5 NFE      | 6NFE     | 8NFE      |  10NFE      |  12NFE      |  15NFE     |  20NFE      |
> |---------------|-------------------|-------------------|-------------------|-------------------|-------------------|-------------------|-------------------|
> | DPM-Solver++ (*baseline*)|15.69              | 11.65              | 9.06              |8.29             |7.94             |7.70              |7.48              |
> | UniPC  |15.03             | 11.30            | 9.07              |8.36             |8.01            |7.71             |7.47             |
> | DPM-Solver-v3  |14.92              | 11.13             | 8.98           |*8.14*            |7.93             |7.70              |7.42              |
> | RE (*our*) |**13.98**              | **10.98**              | **8.84**              |*8.14*            |**7.79**             |**7.48**              |**7.25**            |
>
>
> *4. Can this method be applied to image editing tasks with inversion?*
>
> Honestly, we are not deeply familiar with this specific area. However, our method is compatible with Stable Diffusion. Perhaps techniques from Stable Diffusion could be adapted to image editing tasks involving inversion.
>
>
> We are once again grateful for your encouragement. It gave us a sense of hope, as if we caught a glimpse of light at a particular moment, and we are truly thankful.  We hope these clarifications address your concerns.

---

### Author Response · Authors · 2024-12-03
**Reminder on Reviewer-Author Discussion for Submission #2183**

Dear Reviewers,

We hope this message finds you well. We are writing to kindly remind you of the ongoing reviewer-author discussion for our submission (#2183). We have carefully addressed the concerns raised in your reviews and submitted our rebuttal.

As the discussion deadline is approaching, we would greatly appreciate it if you could share your feedback on whether our responses have adequately addressed your concerns or if there are any remaining points that require clarification.

Thank you all once again for your time and thoughtful efforts during this busy period.

Best regards,

The Authors

---

### Meta-Review · Area_Chair_KGv3 · 2024-12-21

**Metareview:**

Summary. This paper examines the efficiency of denoising diffusion models (DMs) by introducing an entropy-focused approach to improve denoising iterations. It proposes conditional entropy reduction as a means to streamline denoising iterations, leveraging gradient estimation techniques without additional model training.

Strengths.
The paper offers interesting perspective on improving sampling.

Weaknesses.
The motivation of reducing entropy needs additional discussion.
The improvement of performance at fewer than 10 NFEs is still challenging.
The writing of the paper needs some improvement. The mathematical details and derivations can be made more accessible to broad readership.

Missing.
The paper is missing the motivation and background to digest the technical details.
The paper did not provide code, which hinders reproducibility.

Reasons.
I find the paper difficult to understand. While reducing conditional entropy makes intuitive sense, the paper does not provide clear description or empirical results on what aspects of the diffusion process are improved.

**Additional Comments On Reviewer Discussion:**

Reviewers raised concerns about the quality of writing, key assumptions and claims in the paper, lack of visual results.

Authors addressed some of the concerns and modified several parts of the text.
Overall, reviewers acknowledge that the idea of using conditional entropy is interesting, but the paper needs a major revision. AC agrees with that.

---

### Decision · Program_Chairs · 2025-01-22

Reject